# Learning to Cooperate under Private Rewards

## Abstract

We address a critical challenge in multi-agent reinforcement learning (MARL): maximizing team rewards in scenarios where agents only have access to their individual, private rewards. This setting presents unique challenges, as agents need to cooperate to optimize collective performance whilst having only local, potentially conflicting objectives. Existing MARL methods often tackle this by sharing rewards, values, or full policies, but these approaches raise concerns about privacy and computational overhead. We introduce Anticipation Sharing (AS), a novel MARL method that achieves team-level coordination through the exchange of anticipated peer action distributions. Our key theoretical contribution is a proof that the deviation between the collective return and individual objectives can be identified through these anticipations. This allows AS to align agent behaviours towards team objectives without compromising individual privacy or incurring the prohibitive costs of full policy sharing. Experimental results demonstrate that AS is competitive with baseline algorithms that share values or policy parameters, whilst offering significant advantages in privacy preservation and computational efficiency. Our work presents a promising direction for reward-private cooperative MARL in scenarios where agents must maximize team performance using only their private, individual rewards.

## 1 Introduction

Multi-agent reinforcement learning (MARL) enables collaborative decision-making in diverse real-world applications, such as autonomous vehicles (Xia et al., 2022; Qiu et al., 2023), robotics (Wang et al., 2022; Peng et al., 2021; Sun et al., 2020), and communications systems (Siedler & Alpha; Huang & Zhou, 2022). A key open challenge in this field is enabling agents to maximize team rewards while having access to only their private and potentially conflicting individual rewards. This need commonly arises in many practical scenarios, particularly in social dilemmas, where optimizing rewards based solely on individual interests often leads to suboptimal collective outcomes and, consequently, suboptimal individual outcomes. For instance, in distributed energy systems like a smart grid, various energy producers and consumers need to coordinate to balance supply and demand. Each entity has its own cost/utility function (individual reward) that it may not want to disclose. Yet, the stability and efficiency of the entire grid (collective return) depend on their coordinated actions. If each entity maximizes its own utility, the entire stability might be degraded and thus the individual entity will only achieve a suboptimal outcome. In supply chain optimization, multiple companies need to coordinate their production and logistics to maximize overall efficiency. Each company has its own profit function (individual reward) that it wants to keep private due to competitive concerns. However, the overall supply chain performance (collective return) depends on their coordinated actions. Additionally, in smart healthcare, multiple hospitals aim to collaboratively train a medical diagnosis model without sharing patient data. Each hospital has its own performance metric (individual reward) based on its specific patient population and priorities. A problem is to create a model that performs well across all hospitals (collective return) without compromising individual hospital data or metrics. These scenarios underline the critical need for MARL strategies that enable agents to effectively learn cooperative behaviors, despite operating with only their private, individual rewards.

The issue of privacy concerning rewards, values, and policies presents a significant hurdle in these scenarios (Xu et al., 2021; Yuan et al., 2023; Ma et al., 2023). Agents often prioritise the confidentiality of their policies,

rewards, and values to mitigate risks such as malicious attacks, unwanted disclosure of strategic interests, and potential loss of intellectual property. These privacy concerns introduce unique challenges, as agents are required to collaboratively optimize collective performance while only having access to local and potentially conflicting objectives, and without the capability to directly share sensitive information.

Numerous MARL methods have been proposed to enable cooperation under an evenly split shared reward to each agent (Sun et al., 2022; Lauer & Riedmiller, 2000; Boutilier, 1996; Jiang & Lu, 2022; Kuba et al., 2022; Wu et al., 2021). However, these methods are infeasible when an agent is privy only to its individual reward, as in our setup. Under individual rewards, agents may face social dilemmas when they have conflicting interests - prioritising individual rewards can produce suboptimal collective outcomes. The Prisoner's Dilemma exemplifies this tension. When agents act purely out of self-interest, they will choose to defect, which leads to lower total returns compared to cooperating for the common good (Debreu, 1954). However, determining optimal collaborative strategies is challenging when each agent only sees a local viewpoint.

To enhance cooperation towards maximizing the collective interest under individual rewards, several methods have been proposed. These approaches acknowledge the challenge of using only individual rewards, but typically assume that potentially private information can be shared across agents. Some strategies involve sharing rewards to guide agents towards a collective optimum (Chu et al., 2020b; Yi et al., 2022; Chu et al., 2020a). Others suggest sharing the model parameters of the value functions or the output value of the value functions, and through aggregation from neighbouring agents, they guide agents to achieve collective optimum (Zhang et al., 2018a;b; 2020; Suttle et al., 2020; Du et al., 2022). In these approaches, agents calculate a global value based on shared rewards or values, and subsequently, they adjust their policies to maximize this global value. Other studies have explored consensus strategies focusing on policy model parameter sharing rather than values (Zhang & Zavlanos, 2019; Stankovic et al., 2022a;b). While these methods have shown promise in maximizing team rewards, they all rely on the assumption that agents can freely exchange potentially sensitive information. Additionally, sharing model parameters incurs substantial communication overhead, which may transfer excessive and non-essential information, thereby slowing the learning process.

To overcome these challenges, we introduce a novel approach called *Anticipation Sharing* (AS) for cooperative policy learning towards maximizing the total return when agents have individual rewards. A key advantage of our method is achieving emergent collaboration without sharing sensitive information like rewards or model parameters between agents, addressing the limitations of existing methods that rely on such sharing. The core concept we leverage in AS is the exchange of anticipated action distributions, which reflect agents' preferences regarding others. These anticipations are determined by each agent to maximize an individual objective and then sent to corresponding agents for inclusion as constraints when maximizing their objectives. Such anticipations implicitly carry information about individual returns while preserving privacy. By exchanging anticipations, agents can estimate their impacts on collective return without directly sharing sensitive information.

We make a significant theoretical contribution by establishing a lower bound that quantifies the discrepancy between the collective return and the total of individual objectives concerning the anticipated policies. This insight enables the formulation of a surrogate objective for each agent, which is aligned with the global goal and only dependent on local information. Our algorithm optimizes this surrogate objective through a dual-clipped policy update approach, which imposes constraints that penalize deviations between an agent's actual policy and the anticipated ones from its peers. This drives agents to iteratively and distributively move toward collectively optimal policies, addressing the challenge of aligning individual agent policies with the collective optimum. Our empirical results reinforce the validity and practical effectiveness of the AS framework, demonstrating its competitive performance compared to traditional methods based on sharing values or policies across a range of cooperative MARL tasks. This establishes AS as both a theoretically sound and practically effective approach for achieving cooperation in multi-agent systems under individual rewards.

The rest of this paper is structured as follows: Section 2 discusses related work on cooperative MARL under individual rewards. Section 3 presents the technical background and problem formulation. Section 4 introduces our Anticipation Sharing methodology, including the theoretical foundations and practical

algorithm. Section 5 details our experimental setup and results. Finally, Section 6 concludes with a discussion of the implications of our work and future research directions.

## 2 Related work

In this work, we focus on cooperative MARL under individual reward, which is distinguished from numerous contemporary studies that focus on optimizing multi-agent policies under the assumption of an evenly split shared team reward (Kuba et al., 2022; Wu et al., 2021; Sun et al., 2022; Jiang & Lu, 2022). Cooperation under individual rewards reflects a more realistic scenario in many real-world applications, where agents need to learn to cooperate based on limited and individual information due to privacy or scalability concerns.

With individual reward setup, many works Lowe et al. (2017); Iqbal & Sha (2019); Foerster et al. (2017); Omidshafiei et al. (2017); Kim et al. (2021); Jaques et al. (2019) focus on solving Nash equilibrium of a Markov game, i.e., agent seeks the policy that maximizes its own expected return. However, that may not result in collective optimum when agents have conflicting individual interests that can hinder collective cooperation such as in social dilemma. Our research focuses on maximizing the total return across all agents where each agent needs to cooperate to achieve collective optimum. In the rest of this section, we introduce related works aiming to solve this problem.

**Social dilemmas.** Social dilemmas highlight the tension between individual pursuits and collective outcomes. In these scenarios, agents aiming for personal gains can lead to compromised group results. For instance, one study has explored self-driven learners in sequential social dilemmas using independent deep Q-learning (Leibo et al., 2017). A prevalent research direction introduces intrinsic rewards to encourage collective-focused policies. For example, *moral learners* have been introduced with varying intrinsic rewards (Tennant et al., 2023) whilst other approaches have adopted an inequity aversion-based intrinsic reward (Hughes et al., 2018) or rewards accounting for social influences and predicting other agents' actions (Jaques et al., 2019). Borrowing from economics, a method integrated formal contracting to motivate global collaboration (Christoffersen et al., 2023). While these methods modify foundational rewards, we maintain original rewards, emphasizing a collaborative, information-sharing strategy to nurture cooperative agents.

**Value sharing**. Value sharing methods use shared Q-values or state-values among agents to better align individual and collective goals. Many of these methods utilize consensus techniques to estimate the value of a joint policy and guide individual policy updates accordingly. For instance, a number of networked actor-critic algorithms exist based on value function consensus, wherein agents merge individual value functions towards a global consensus by sharing parameters (Zhang et al., 2018a;b; 2020; Suttle et al., 2020). Instead of sharing value function parameters, (Du et al., 2022) shares function values for global value estimation. However, these methods have an inherent limitation: agents modify policies individually, using fixed Q-values or state-values, making them less adaptive to immediate policy shifts from peers, which may introduce policy discoordination. In contrast, our approach enables more adaptive coordination by having agents directly share and respond to peer anticipations.

**Reward sharing**. Reward sharing is about receiving feedback from a broader system-wise outcome perspective, ensuring that agents act in the group's collective best interest. Some works have introduced a spatially discounted reward function (Chu et al., 2020b;a). In these approaches, each agent collaboratively shares rewards within its vicinity. Subsequently, an adjusted reward is derived by amalgamating the rewards of proximate agents, with distance-based discounted weights. Other methods advocate for the dynamic learning of weights integral to reward sharing, which concurrently evolve as agents refine their policies (Yi et al., 2022). In our research, we focus on scenarios where agents know only their individual rewards and are unaware of their peers' rewards. This mirrors real-world situations where rewards are kept confidential or sharing rewards suffers challenges such as communication delays and errors. Consequently, traditional value or reward sharing methods fall short in these contexts. In contrast, our method induces coordination without requiring reward sharing.

**Policy sharing**. Policy sharing strives to unify agents' behaviors through an approximate joint policy. However, crafting a global policy for each agent based on its individual reward can lead to suboptimal outcomes. Consensus update methods offer a solution by merging individually learned joint policies towards

an optimal joint policy. Several studies have employed such a strategy, focusing on a weighted sum of neighboring agents' policy model parameters (Zhang & Zavlanos, 2019; Stankovic et al., 2022a;b). These methods are particularly useful when sharing individual rewards or value estimates is impractical. Yet, sharing policy model parameters risks added communication overheads and data privacy breaches. PS is based on the idea of federated learning and shares the parameters of joint policies among agents. In contrast, our method focuses on learning individual policies and sharing only the relevant action distributions of the anticipated policies with the corresponding agents, which typically involves less communication overhead compared to sharing entire policy parameters with all the neighbouring agents.

**Teammate modeling** Teammate/opponent modeling in MARL often relies on agents having access to, or inferring, information about teammates' goals, actions, or rewards. This information is then used to improve collective outcomes (Albrecht & Stone, 2018; He et al., 2016; Wen et al., 2019; Zheng et al., 2018). Our approach differs from traditional team modeling. Rather than focusing on predicting teammates' exact actions or strategies, our method has each agent calculate and share anticipated action distributions that would benefit its own strategy. These anticipations are used by other agents (not the agent itself) to balance their objectives with that of the agent sending the anticipation. This approach emphasizes anticipations that serve the agent's own objective optimization. Coordination occurs through policy adaptation based on *others' anticipations* that implicitly include information about their returns, rather than modeling their behaviors. It contrasts with conventional team modeling in MARL that focuses on modeling teammates' behaviors directly.

## 3 Background and problem statement

We approach the collaborative MARL problem with individual rewards using a Multi-agent Markov Decision Process (MMDP), which was also employed in previous works. Zhao et al. (2020); Krouka et al. (2022) formalized the same problem as we did. Chen et al. (2022) considered a similar problem, but with a central controller that can collect information from all agents. Zhang et al. (2018b); Du et al. (2022); Sha et al. (2021) used the same basic problem formalism, but added a network structure on agent systems, referring to it as Networked MMDP or MARL over networks. Additionally, Lei et al. (2022) presented the Networked MARL problem from the perspective of Alternating Direction Method of Multipliers (ADMM).

Specifically, we consider an MMDP with $N$ agents represented as a tuple $< \mathcal{S}, \{\mathcal{A}^i\}_{i=1}^N, \mathcal{P}, \{\mathcal{R}^i\}_{i=1}^N, \gamma >$, where $\mathcal{S}$ denotes a global state space, $\mathcal{A}^i$ is the individual action space, $\mathcal{A} = \Pi_{i=1}^N \mathcal{A}^i$ is the joint action space, $\mathcal{P} : \mathcal{S} \times \mathcal{A} \times \mathcal{S} \to [0, 1]$ is the state transition function, $\mathcal{R}^i : \mathcal{S} \times \mathcal{A} \to \mathbb{R}$ is the individual reward function, and $\gamma$ is a discount factor. Each agent $i$ selects action $a^i \in \mathcal{A}^i$ based on its individual policy $\pi^i : \mathcal{S} \times \mathcal{A}^i \to [0, 1]$. The joint action of all agents is represented by $\boldsymbol{a} \in \mathcal{A}$, and the joint policy across these agents is denoted as $\boldsymbol{\pi}(\cdot|s) = \prod_{i=1}^N \pi^i(\cdot|s)$. The objective is to maximize the expectation of collective cumulative return of all agents,

$$\eta(\boldsymbol{\pi}) = \sum_{i=1}^N \mathbb{E}_{\tau \sim \boldsymbol{\pi}} \left[ \sum_{t=0}^\infty \gamma^t r_t^i \right], \tag{1}$$

where the expectation, $\mathbb{E}_{\tau \sim \boldsymbol{\pi}}[\cdot]$, is computed over trajectories with an initial state distribution $s_0 \sim d(s_0)$, action selection $\boldsymbol{a}_t \sim \boldsymbol{\pi}(\cdot|s_t)$, state transitions $s_{t+1} \sim \mathcal{P}(\cdot|s_t, \boldsymbol{a}_t)$, and $r_t^i = \mathcal{R}^i(s, \boldsymbol{a})$ is the reward for individual agent $i$. Here we use $r_t^i = R^i(s, a)$ for simplicity of notation, but this can be easily extended to a stochastic reward function without affecting the core of our method. An individual advantage function is defined as:

$$A_i^{\boldsymbol{\pi}}(s, \boldsymbol{a}) = Q_i^{\boldsymbol{\pi}}(s, \boldsymbol{a}) - V_i^{\boldsymbol{\pi}}(s) \tag{2}$$

which depends on the individual state-value and action-value functions, respectively,

$$V_i^{\boldsymbol{\pi}}(s) = \mathbb{E}_{\tau \sim \boldsymbol{\pi}} \left[ \sum_{t=0}^\infty \gamma^t r_t^i | s_0 = s \right], \quad Q_i^{\boldsymbol{\pi}}(s, \boldsymbol{a}) = \mathbb{E}_{\tau \sim \boldsymbol{\pi}} \left[ \sum_{t=0}^\infty \gamma^t r_t^i | s_0 = s, \boldsymbol{a}_0 = \boldsymbol{a} \right]. \tag{3}$$

Our problem setup is similar to Stochastic Games (Markov Games) (Shapley, 1953) in terms of structure, and to Dec-POMDP (Bernstein et al., 2002) in terms of the optimization objective. However, there are key distinctions. Unlike standard Stochastic Games, our agents are cooperative and aim to maximize a collective return. Unlike Dec-POMDP, our agents have access to the full state (not partial observations) and individual reward functions (not a shared reward signal). In our setup, agents do not have direct access to others'

policies, rewards, or values. This setting is particularly relevant for applications where privacy concerns or decentralized control are important. It enables us to explore the balance between cooperative behavior and individual privacy in multi-agent systems, which is crucial in many real-world applications. With the private individual rewards, an agent naïvely optimizing its individual reward might take actions that are suboptimal for the group. Our work aims to bridge this gap between individual reward optimization and collective return maximization. It enables agents to approximate the optimization of the collective objective while operating solely with their individual reward signals. In the next section, we present a method where agents iteratively share anticipations to maximize a lower bound of Eq.1. This method is general and not dependent on any specific protocol for communicating anticipations between agents. In Sec.4.3, we propose a practical algorithm that involves sharing information within agents' neighbourhoods. Our experiments demonstrate the effects of different sharing protocols on the performance of MARL cooperation.

## 4 Methodology

In cooperative MARL settings with individual rewards, agents must balance personal objectives with collective goals, despite lacking global perspectives. Our approach, *anticipation sharing* (AS), facilitates this dual awareness without direct reward or objective sharing. In AS, agents exchange anticipations about peer actions, which are derived by maximizing their own objectives. These anticipations are then considered by other agents when maximizing their individual objectives, enabling each agent to infer the collective objective.

Unlike traditional methods that share explicit rewards or objectives, AS involves agents exchanging anticipations that implicitly contain information about others' objectives. By observing how its actions align with aggregated anticipations, each agent can perceive the divergence between its individual interests and the inferred collective goals. This drives policy updates to reduce the identified discrepancy, bringing local and global objectives into closer alignment.

Our approach leverages these identified divergences to align agents' policies. Through iterative sharing of anticipated actions and policy adaptation, AS fosters continuous, adaptive refinement of strategies that balance both individual objectives and the collective goal.

### 4.1 Theoretical developments

We commence our technical developments by analyzing joint policy shifts based on global information. This extends foundational trust region policy optimization work (Schulman et al., 2015) to multi-agent settings with individual advantage values, which is distinguished from previous works (Wu et al., 2021; Su & Lu, 2022) that are based on common rewards and advantages. We prove the following bound on the expected return difference between new and old joint policies:

**Lemma 1** *We establish a bound for the difference in expected returns between an old joint policy $\boldsymbol{\pi}_{old}$ and a newer policy $\boldsymbol{\pi}_{new}$:*

$$\eta(\boldsymbol{\pi}_{new}) \geq \eta(\boldsymbol{\pi}_{old}) + \zeta_{\boldsymbol{\pi}_{old}}(\boldsymbol{\pi}_{new}) - C \cdot D_{KL}^{max}(\boldsymbol{\pi}_{old}||\boldsymbol{\pi}_{new}), \tag{4}$$

*where*

$$\zeta_{\boldsymbol{\pi}_{old}}(\boldsymbol{\pi}_{new}) = \mathbb{E}_{s \sim d^{\boldsymbol{\pi}_{old}}(s), \boldsymbol{a} \sim \boldsymbol{\pi}_{new}(|s)} \left[ \sum_i A_i^{\boldsymbol{\pi}_{old}}(s, \boldsymbol{a}) \right], \quad C = \frac{4 \max_{s, \boldsymbol{a}} |\sum_i A_i^{\boldsymbol{\pi}_{old}}(s, \boldsymbol{a})| \gamma}{(1 - \gamma)^2}$$

$$D_{KL}^{max}(\boldsymbol{\pi}_{old}||\boldsymbol{\pi}_{new}) = \max_s D_{KL}(\boldsymbol{\pi}_{old}(\cdot|s)||\boldsymbol{\pi}_{new}(\cdot|s)). \tag{5}$$

The proof is given in Appendix A.1.1.

The key insight is that the improvement in returns under the new policy depends on both the total advantages of all the agents, as well as the divergence between joint policy distributions. This quantifies the impact of joint policy changes on overall system performance given global knowledge, extending trust region concepts to multi-agent domains.

However, as the improvement in returns is measured by joint policy distributions and total advantages of all agents, it is hard to be used by single agent in MARL settings where each agent has no access to others' policies and rewards. To address this limitation, we first introduce the concept of *anticipated joint policy* from each agent's local perspective to replace the true joint policy. As we will show in Sec. 4.2, the anticipated joint policy of each agent is solved by optimizing an individual objective. Analyzing anticipated policies is crucial for assessing the discrepancy between individual objectives and the collective one in cooperative MARL.

**Definition 1** *For each agent in a multi-agent system, we define the **anticipated joint policy**, denoted as $\tilde{\pi}^i$, formulated as $\tilde{\pi}^i(\boldsymbol{a}|s) = \prod_{j=1}^{N} \pi^{ij}(a^j|s)$. Here, for each agent $i$, $\pi^{ij}$ represents the anticipation of agent $i$ about agent $j$'s policy when $j \neq i$. When $j = i$, we have $\pi^{ii} = \pi^i$, which is agent $i$'s own policy. To represent the collection of all such anticipated joint policies across agents, we use the notation $\tilde{\boldsymbol{\Pi}} := (\tilde{\pi}^1, \cdots, \tilde{\pi}^i, \cdots, \tilde{\pi}^N)$.*

The anticipated joint policy represents an agent's perspective of the collective strategy constructed from its own policy and anticipations to peers. We will present how to solve such anticipated joint policy in Sec. 4.2.

**Definition 2** *The total expectation of individual advantages over the anticipated joint policies and a common state distribution, is defined as follows:*

$$\zeta_{\pi'}(\tilde{\boldsymbol{\Pi}}) = \sum_i \mathbb{E}_{s \sim d^{\pi'}(s), \boldsymbol{a} \sim \tilde{\pi}^i(\boldsymbol{a}|s)} \left[ A_i^{\pi'}(s, \boldsymbol{a}) \right], \tag{6}$$

*which represents the sum of expected advantages for each agent $i$, calculated over their anticipated joint policy $\tilde{\pi}^i$ and a shared state distribution, $d^{\pi'}(s)$. The advantage $A_i^{\pi'}(s, \boldsymbol{a})$ for each agent is evaluated under a potential joint policy $\pi'$, which may differ from the true joint policy $\pi$ in play. This definition captures the expected benefit each agent anticipates based on the anticipated joint actions, relative to the potential joint policy $\pi'$.*

This concept quantifies the expected cumulative advantage an agent could hypothetically gain by switching from a reference joint policy to the anticipated joint policies of all agents. It encapsulates the perceived benefit of the anticipated policies versus a collective benchmark. Intuitively, if an agent's anticipations are close to the actual policies of other agents, this expected advantage will closely match the actual gains. However, discrepancies in anticipations will lead to divergences, providing insights into the impacts of imperfect local knowledge.

Equipped with these notions of anticipated joint policies and total advantage expectations, we can analyze the discrepancy of the expectation of the total advantage caused by policy shift from the true joint policy, $\pi$, to the individually anticipated ones, $\tilde{\boldsymbol{\Pi}}$. Specifically, we prove the following bound relating this discrepancy:

**Lemma 2** *The discrepancy between $\zeta_{\pi'}(\tilde{\boldsymbol{\Pi}})$ and $\zeta_{\pi'}(\pi)$ is upper bounded as follows:*

$$\zeta_{\pi'}(\tilde{\boldsymbol{\Pi}}) - \zeta_{\pi'}(\pi) \leq f^{\pi'} + \sum_i \frac{1}{2} \max_{s,\boldsymbol{a}} \left| A_i^{\pi'}(s, \boldsymbol{a}) \right| \cdot \sum_{s,\boldsymbol{a}} \left( \tilde{\pi}^i(\boldsymbol{a}|s) - \pi(\boldsymbol{a}|s) \right)^2, \tag{7}$$

*where*

$$f^{\pi'} = \sum_i \frac{1}{2} \max_{s,\boldsymbol{a}} \left| A_i^{\pi'}(s, \boldsymbol{a}) \right| \cdot |\mathcal{A}| \cdot \|d^{\pi'}\|_2^2, \tag{8}$$

and $\|d^{\pi'}\|_2^2 = \sum_s (d^{\pi'}(s))^2$.

The proof is given in Appendix A.1.2.

This result quantifies the potential drawbacks of relying on imperfect knowledge in cooperative MARL settings, where agents' anticipations may diverge from actual peer policies. It motivates reducing the difference between anticipated and true joint policies.

Previous results bounded the deviation between total advantage expectations under the true joint policy versus under anticipated joint policies. We now build on this to examine how relying too much on past experiences and anticipated joint policies can lead to misjudging the impact of new joint policy shifts over time. To this

end, we consider the relationship between $\zeta_{\boldsymbol{\pi}_{\text{old}}}(\tilde{\boldsymbol{\Pi}}_{\text{new}})$, the perceived benefit of the new anticipated joint policies $\tilde{\boldsymbol{\Pi}}_{\text{new}}$, assessed from the perspective of the previous joint policy $\boldsymbol{\pi}_{\text{old}}$, and $\eta(\boldsymbol{\pi}_{\text{new}})$, which measures the performance of the new joint policy. Specifically, $\zeta_{\boldsymbol{\pi}_{\text{old}}}(\tilde{\boldsymbol{\Pi}}_{\text{new}})$ is defined like Definition 2 as:

$$\zeta_{\boldsymbol{\pi}_{old}}(\tilde{\boldsymbol{\Pi}}_{new}) = \sum_i \mathbb{E}_{s \sim d^{\boldsymbol{\pi}_{old}}(s), \boldsymbol{a} \sim \tilde{\boldsymbol{\pi}}^i_{new}(\boldsymbol{a}|s)} \left[ A_i^{\boldsymbol{\pi}_{old}}(s, \boldsymbol{a}) \right], \tag{9}$$

which represents a potentially myopic and individual perspective informed by the advantage values, $A_i^{\boldsymbol{\pi}_{old}}$, of past policies, as well as individually anticipated joint policies, $\tilde{\boldsymbol{\pi}}^i_{new}$, and thus, it may inaccurately judge the actual impact of switching to $\boldsymbol{\pi}_{\text{new}}$ as quantified by $\eta(\boldsymbol{\pi}_{\text{new}})$. The following theorem provides a lower bound of the deviation between the collective return, $\eta(\boldsymbol{\pi}_{\text{new}})$, of the newer joint policy, and $\zeta_{\boldsymbol{\pi}_{\text{old}}}(\tilde{\boldsymbol{\Pi}}_{\text{new}})$.

**Theorem 1** *The discrepancy between the return of the newer joint policy and the value of $\zeta_{\boldsymbol{\pi}_{old}}(\tilde{\boldsymbol{\Pi}}_{new})$ is lower bounded as follows:*

$$\eta(\boldsymbol{\pi}_{new}) - \zeta_{\boldsymbol{\pi}_{old}}(\tilde{\boldsymbol{\Pi}}_{new}) \geq$$
$$\eta(\boldsymbol{\pi}_{old}) - C \cdot \sum_i D_{KL}^{max}(\pi_{old}^{ii}||\pi_{new}^{ii}) - f^{\boldsymbol{\pi}_{old}} - \sum_i \frac{1}{2} \max_{s, \boldsymbol{a}} \left| A_i^{\boldsymbol{\pi}_{old}}(s, \boldsymbol{a}) \right| \cdot \sum_{s, \boldsymbol{a}} \left( \tilde{\boldsymbol{\pi}}^i_{new}(\boldsymbol{a}|s) - \boldsymbol{\pi}_{new}(\boldsymbol{a}|s) \right)^2. \tag{10}$$

The full proof is given in Appendix A.1.3.

This theorem explains the nuanced dynamics of policy changes in MARL where agents learn separately. It sheds light on how uncoordinated local updates between individual agents affect the collective performance. At the same time, this result suggests a potential way to improve overall performance by leveraging the anticipated joint policies held by each agent.

## 4.2 A surrogate optimization objective

Our preceding results established analytical foundations for assessing joint policy improvement in multi-agent settings with individual rewards. We now build upon these results to address the practical challenge of optimizing system-wide returns when agents lack knowledge of others' policies, rewards, and values.

Directly maximizing the expected collective returns, $\eta(\boldsymbol{\pi})$, is intractable without global knowledge of the joint policy and collective return. However, Theorem 1 provides insight into a more tractable approach: agents can optimize a localized surrogate objective, $\zeta_{\boldsymbol{\pi}_{\text{old}}}(\tilde{\boldsymbol{\Pi}})$, which is the sum of individual objectives concerning anticipated joint policies and individual advantage values. This simplifies the global objective into an individual form dependent on the anticipated joint policy that is composed of an agent's individual policy, $\pi^{ii}$, and its anticipations of others, $\pi^{ij}$.

To leverage this insight, we use the lower bound given by Theorem 1. By maximizing this lower bound plus $\zeta_{\boldsymbol{\pi}_{\text{old}}}(\tilde{\boldsymbol{\Pi}})$, we can maximize the collective return. We can ignore the terms $\eta(\boldsymbol{\pi}_{\text{old}})$ and $f^{\boldsymbol{\pi}_{\text{old}}}$ from Theorem 1 in our optimization problem, as they are not relevant to optimizing $\tilde{\boldsymbol{\Pi}}$ and their values are usually bounded. To be specific, the value of $\eta(\boldsymbol{\pi}_{\text{old}})$ is bounded as the reward value is bounded. For $f^{\boldsymbol{\pi}_{\text{old}}}$, as defined in Eq. 8, its value is also bounded since (1) We focus on scenarios with finite and relatively small action spaces, which are common in many real-world applications, so $|\mathcal{A}|$ (the size of the action space) is not excessively large. (2) The term $\|d^{\pi_{\text{old}}}\|_2^2$ is the squared L2-norm of the state visitation distribution, which is bounded.(3) The advantage function $A_i^{\pi_{\text{old}}}(s, a)$ is also bounded as the reward value is bounded.

Consequently, we propose the following global constrained optimization problem as a surrogate for the original collective objective:

$$\max_{\tilde{\boldsymbol{\Pi}}} \sum_i \mathbb{E}_{s \sim d^{\boldsymbol{\pi}_{old}}(s), \boldsymbol{a} \sim \tilde{\boldsymbol{\pi}}^i(\boldsymbol{a}|s)} \left[ A_i^{\boldsymbol{\pi}_{old}}(s, \boldsymbol{a}) \right]$$
$$\text{s.t.} \quad \sum_i D_{KL}^{max}(\pi_{old}^{ii}||\pi^{ii}) \leq \delta, \qquad \sum_i \max_{s, \boldsymbol{a}} \left| A_i^{\boldsymbol{\pi}_{old}}(s, \boldsymbol{a}) \right| \cdot \sum_{s, \boldsymbol{a}} \left( \tilde{\boldsymbol{\pi}}^i(\boldsymbol{a}|s) - \boldsymbol{\pi}(\boldsymbol{a}|s) \right)^2 \leq \delta'. \tag{11}$$

Note that, taking into account of the results given by (Schulman et al., 2015), we do not directly include the lower bound of the discrepancy given by Eq. 10 in Eq. 11, but instead use constraints to facilitate learning.

Eq. 11 captures the essence of coordinating joint policies to maximize localized advantages with anticipated joint policies. However, it still assumes full knowledge of $\tilde{\mathbf{\Pi}}$. To make this feasible in individual policy learning, we reformulate it from each agent's perspective. Remarkably, we can distill the relevant components into a local objective and constraints for each individual agent $i$, as follows:

$$
\max_{\tilde{\boldsymbol{\pi}}^i} \mathbb{E}_{s \sim d^{\boldsymbol{\pi}_{old}(s)}, \boldsymbol{a} \sim \tilde{\pi}^i(\boldsymbol{a}|s)} \left[ A_i^{\boldsymbol{\pi}_{old}}(s, \boldsymbol{a}) \right]
$$
$$
\text{s.t.:} \quad \text{(a)} \quad D_{KL}^{max}(\pi_{old}^{ii}||\pi^{ii}) \leq \delta_1, \quad \text{(b)} \quad \kappa_i \cdot \sum_{s, a_j} (\pi^{ij}(a_j|s) - \pi^{jj}(a_j|s))^2 \leq \delta_2, \ \forall j \neq i,
$$
$$
\text{(c)} \quad \kappa_i \cdot \sum_{s, a_i} (\pi^{ii}(a_i|s) - \pi^{ji}(a_i|s))^2 \leq \delta_2, \ \forall j \neq i,
$$
(12)

where $\kappa_i = \max_{s, \boldsymbol{a}} |A_i^{\boldsymbol{\pi}_{old}}(s, \boldsymbol{a})|$.

The constraints in Eq. 12 are imposed on $\pi^{ii}$ and $\pi^{ij}$ ($j \neq i$), which together compose $\tilde{\boldsymbol{\pi}}^i$. Therefore, these constraints effectively limit the space of possible $\tilde{\boldsymbol{\pi}}^i$ by constraining its components. Constraint (a) limits how much the agent's own policy can change, while constraints (b) and (c) ensure that the anticipations are close to the actual policies of other agents. The constraints also depend on other agents' policies $\pi^{jj}$ and their anticipations of agent $i$'s policy, $\pi^{ji}$. To evaluate these terms, each agent $j$ shares its action distribution $\pi^{jj}(\cdot|s)$ and the anticipated action distribution $\pi^{ji}(\cdot|s)$ with agent $i$. This sharing enables each agent $i$ to assess the constraint terms, which couple individual advantage optimizations under local constraints. These constraints reflect both the differences between the true policies of others and an agent's anticipations of them, as well as the discrepancy between an agent's own true policy and others' anticipations of it. By distributing the optimization while exchanging critical policy information, this approach balances individual policy updates while maintaining global coordination among agents.

It's important to distinguish our anticipated policy learning objective from traditional teammate modeling. In teammate modeling, agent $i$ typically approximates peer policies $\hat{\pi}^{ij}$ and uses these approximations when solving for its own policy $\pi^{ii}$. In contrast, our approach in Eq. 12 aims to optimize the anticipations $\pi^{ij}$ alongside $\pi^{ii}$. These optimized anticipations $\pi^{ij}$ are then used by agent $j$ to solve for its policy $\pi^{jj}$. This method allows the anticipations to implicitly incorporate information about individual objectives. Through the exchange of these anticipations, individual agents can balance others' objectives and, consequently, the collective performance while optimizing their own objectives.

### 4.3  A practical algorithm for learning with AS

We propose a structured approach to optimize the objective in Eq. 12. The derivation of the algorithm involves specific steps, each targeting different aspects of the optimization challenge. Note that in this practical algorithm, we present a setup where agent $i$ exchanges information with neighbours $\{j|j \in \mathcal{N}_i\}$ that may not include all other $(N-1)$ agents, and is not subject to a particular protocol used for determining $\mathcal{N}_i$. In experiments, we use different neighbourhood definitions/protocols to investigate corresponding effects.

**Step 1: Clipping Policy Ratio for KL Constraint.**  Addressing the KL divergence constraint (a) in Eq. 12 is crucial in ensuring each agent's policy learning process remains effective. This constraint ensures that updates to an agent's individual policy do not deviate excessively from its previous policy. To manage this, we incorporate a clipping mechanism, inspired by PPO-style clipping (Schulman et al., 2017), adapted for individual agents in our method.

We start by defining probability ratios for the individual policy and anticipated peer policies:

$$
\xi_i = \frac{\pi^{ii}(a_i|s'; \theta^{ii})}{\pi_{old}^{ii}(a_i|s'; \theta_{old}^{ii})}, \quad \xi_{\mathcal{N}_i} = \prod_{j \in \mathcal{N}_i} \frac{\pi^{ij}(a_j|s; \theta^{ij})}{\pi_{old}^{jj}(a_j|s; \theta_{old}^{jj})}.
$$
(13)

These ratios measure the extent of change in an agent's policy relative to its previous one and its anticipations to others. We then apply a clipping operation to $\xi_i$, the individual policy ratio:

$$
\mathbb{E}_{s \sim d^{\boldsymbol{\pi}_{old}(s)}, \boldsymbol{a} \sim \pi_{old}(\boldsymbol{a}|s)} \left[ \min \left( \xi_i \xi_{\mathcal{N}_i} \hat{A}_i, \text{clip}(\xi_i, 1 - \epsilon, 1 + \epsilon) \xi_{\mathcal{N}_i} \hat{A}_i \right) \right].
$$

This method selectively restricts major changes to the individual policy $\pi^{ii}$, while allowing more flexibility in updating anticipations of peer policies. It balances the adherence to the KL constraint with the flexibility needed for effective learning and adaptation in a multi-agent environment.

**Step 2: Penalizing Anticipation Discrepancies.** The objective of this step is to enforce constraints (b) and (c) in Eq. 12, which aim to penalize discrepancies between the anticipated and true policies. Simply optimizing the advantage function may not sufficiently increase these discrepancies. To be specific, if $\hat{A}_i > 0$, according to the main objective function, Eq. 12, the gradient used to update $\pi^{ij}$ will be positive and will lead to the increase of $\pi^{ij}$. If $\frac{\pi^{ij}(a|s,\theta^{ij})}{\pi^{jj}(a|s)} < 1$, i.e. $\pi^{ij}(a|s,\theta^{ij}) < \pi^{jj}(a|s)$, then the gradient caused by the main objective will decrease the discrepancy between $\pi^{ij}$ and $\pi^{jj}$. Therefore, we introduce penalty terms that are activated when policy updates inadvertently increase these discrepancies. Specifically, we define state-action sets $X^{ij}$ to identify where the policy update driven by the advantage exacerbates the discrepancies between the resulting anticipated policies and other agents' current policies, and $X^{ii}$ to identify the discrepancies between the resulting agent's own policy and the ones anticipated from other agents. These are defined as:

$$X^{ij} = \left\{ (s, \boldsymbol{a}) \mid \frac{\pi^{ij}(a_j|s;\theta^{ij})}{\pi^{jj}(a_j|s)} \hat{A}_i \geq \hat{A}_i \right\} \qquad X^{ii} = \left\{ (s, \boldsymbol{a}) \mid \frac{\pi^{ii}(a_i|s;\theta^{ii})}{\pi^{ji}(a_i|s)} \hat{A}_i \geq \hat{A}_i \right\}, \tag{14}$$

where the pairs $(s, \boldsymbol{a})$ represent scenarios in which the gradient influenced by $\hat{A}_i$ increases the divergence between the two policies. The following indicator function captures this effect:

$$\mathbb{I}_X(s, \boldsymbol{a}) = \begin{cases} 1 & \text{if } (s, \boldsymbol{a}) \in X, \\ 0 & \text{otherwise.} \end{cases} \tag{15}$$

**Step 3: Dual Clipped Objective.** In the final step, we combine the clipped surrogate objective with coordination penalties to form our dual clipped objective:

$$\max_{\theta^{ii}, \boldsymbol{\theta}_{-ii}} \mathbb{E}_{s \sim d^{\boldsymbol{\pi}_{old}}(s), \boldsymbol{a} \sim \boldsymbol{\pi}_{old}(\boldsymbol{a}|s)} \Big[ \min \big( \xi_i \xi_{\mathcal{N}_i} \hat{A}_i, \text{clip}(\xi_i, 1 - \epsilon, 1 + \epsilon) \xi_{\mathcal{N}_i} \hat{A}_i \big)$$
$$- \kappa_i \cdot \sum_{j \in \mathcal{N}_i} \rho_j \mathbb{I}_{X^{ij}}(s, \boldsymbol{a}) \| \pi^{ij}(\cdot|s; \theta^{ij}) - \pi^{jj}(\cdot|s) \|_2^2 + \rho'_j \mathbb{I}_{X^{ii}}(s, \boldsymbol{a}) \| \pi^{ii}(\cdot|s; \theta^{ii}) - \pi^{ji}(\cdot|s) \|_2^2 \Big], \tag{16}$$

where $\theta^{ii}$ denotes the parameters of $\pi^{ii}$ and $\boldsymbol{\theta}_{-ii}$ denotes the parameters of all the $\pi^{ij}$ ($j \in \mathcal{N}i$). With this objective, each agent optimizes its own policy $\pi^{ii}$ under the constraint of staying close to the anticipated policies. In the meanwhile, the anticipations $\pi^{ij}$ which are involved in $\xi_{\mathcal{N}_i}$, are optimized to maximize the agent's individual advantage function $A_i$ under the constraint of avoiding deviating too far from the actual policies of other agents. This objective function balances individual policy updates with the need for coordination among agents, thereby aligning individual objectives with collective goals. Fig. 1 shows an illustration of our method.

**Implementation details.** In our implementation, we use $\hat{\kappa}_i = \text{mean}_{s,\boldsymbol{a}} |\hat{A}_i^{\boldsymbol{\pi}}|$ to approximate $\kappa_i$ in order to mitigate the impact of value overestimation. Additionally, we adopt the same value for the coefficients $\rho_j$ and $\rho'_j$ across different $j$, and denote it as $\rho$. We also utilize the generalized advantage estimator (GAE) (Schulman et al., 2016) due to its well-known properties to obtain estimates,

$$\hat{A}_i^t = \sum_{l=0}^{\infty} (\gamma \lambda)^l \delta_{t+l}^{V_i}, \qquad \delta_{t+l}^{V_i} = r_i^{t+l} + \gamma V_i(s_{t+l+1}) - V_i(s_{t+l}), \tag{17}$$

where $V_i$ is approximated by minimizing the following loss,

$$\mathcal{L}_{V_i} = \mathbb{E}[(V_i(s_t) - \sum_{l=0}^{\infty} \gamma^l r_i^{t+l})^2]. \tag{18}$$

Algorithm 1 presents the detailed procedure used in our experimental section.

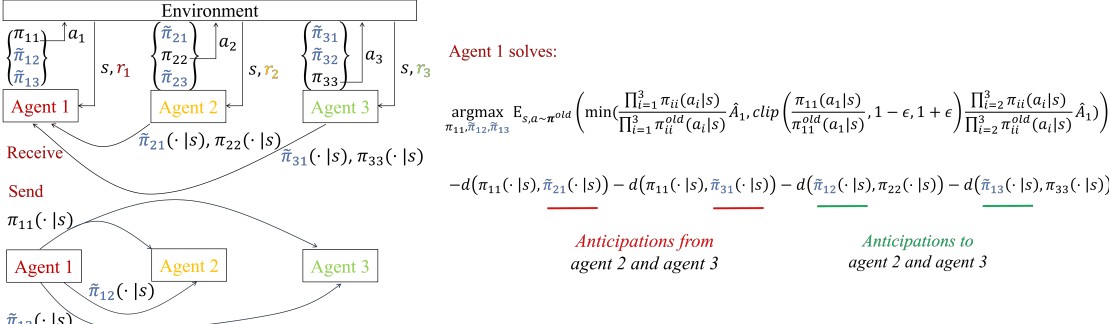

Figure 1: Illustration of AS algorithm, where $d$ represents the function regarding the discrepancy term used in Eq. 16.

---

**Algorithm 1:** Cooperative MARL based on Anticipation Sharing (AS)

---

**Initialize**: Policy networks $\tilde{\boldsymbol{\pi}}^i = (\pi^{i1}, \cdots, \pi^{iN})$, value networks $V_i, \forall i \in \{1, \cdots, N\}$
**for** episode = 1 to $E$ **do**
    $\mathcal{D}_i \leftarrow \phi, \forall i$
    Observe initial state $s_1$
    **for** t=1 to $T$ **do**
        Execute action $a_t^i \in \pi^{ii}(\cdot|s_t)$
        Observe reward $r_t^i$ and next state $s_{t+1}$
        Store $(s_t, a_t^i, r_t^i, s_{t+1}) \in \mathcal{D}_i$
    **end for**
    **for** iteration = 1 to $K$ **do**
        **for** each agent $i$ **do**
            Share action distributions $[\pi_{old}^{ii}(\cdot|s_1), \cdots, \pi_{old}^{ii}(\cdot|s_T)]$ to neighbors $\{j \in \mathcal{N}_i\}$
            Share anticipated action distributions $[\pi_{old}^{ij}(\cdot|s_1), \cdots, \pi_{old}^{ij}(\cdot|s_T)]$ to neighbors $\{j \in \mathcal{N}_i\}$
        **end for**
        **for** i=1 to $N$ **do**
            Compute advantage estimates $\hat{A}_i^1, \cdots, \hat{A}_i^T$ using Eq 17
            Update $\tilde{\boldsymbol{\pi}}^i$ using Eq 16
            Update $V_i$ using Eq 18
            $\tilde{\boldsymbol{\pi}}_{old}^i \leftarrow \tilde{\boldsymbol{\pi}}^i$
            Share action distributions $[\pi_{old}^{ii}(\cdot|s_1), \cdots, \pi_{old}^{ii}(\cdot|s_T)]$ to neighbors $\{j \in \mathcal{N}_i\}$
            Share anticipated action distributions $[\pi_{old}^{ij}(\cdot|s_1), \cdots, \pi_{old}^{ij}(\cdot|s_T)]$ to neighbors $\{j \in \mathcal{N}_i\}$
        **end for**
    **end for**
**end for**

---

# 5 Experiments

## 5.1 Environments

We evaluate our method with five diverse environments where agents have conflicting individual rewards. Three environments are based on related works, while we propose two of our own environments to facilitate the analysis of the problem and the performance of our method.

**Cleanup**. Based on (Christoffersen et al., 2023), this task involves agents cleaning a river and eating apples. Apples spawn only if the waste density of the river is below a threshold. The spawn rate is inversely proportional to the waste density. Eating an apple earns +1 reward for an agent, while cleaning the river yields no reward or cost. This setup creates a free-rider problem - agents may prefer eating apples over cleaning the river, potentially harming collective performance. We set the time horizon of an episode as 100 time steps and environment size as $11 \times 18$.

**Harvest**. Also based on (Christoffersen et al., 2023), this task involves agents harvesting apples. Apples spawn at a rate proportional to the number of apples around the spawn positions. Only eating an apple yields a non-zero reward of +1. The challenge for agents is to harvest apples at a sustainable rate while

collaborating to avoid over-harvesting in the same region. We set the time horizon of an episode as 100 time steps and environment size as $7 \times 38$.

**Cooperative navigation (C. Navigation)**. Building on (Zhang et al., 2018b), this task requires each agent to approach a landmark. We adopt the same observation and action configurations as in (Zhang et al., 2018b). Agents earn rewards based on their proximity to targets but incur a -1 penalty upon collisions. Agents can only exchange information with adjacent counterparts. We set the time horizon of an episode as 100 time steps and use three agents. The environment size is $5 \times 5$. Fig.8(a) in Appendix A.2 illustrates this environment setup.

**Exchange**. We introduce a new discrete environment to explore the dynamics of agent interactions with conflicting interests. Three agents interact with three boxes, each containing food for a specific agent. However, a box can only be opened by another agent at a cost of 5 reward points. The food's intended recipient gains 10 reward points. Every agent loses 0.01 reward points per time step, with a maximum of 300 steps per episode. In a purely self-interested scenario, agents would avoid opening boxes for others, leading to a free-rider problem. The time horizon of an episode is set as 300. The environment size is $5 \times 5$. Appendix A.2, Fig.8(b) illustrates this environment. The state comprises the positions of agents and food items, randomly initialized each episode. Agents can move in four directions, open a box, or stay still. Neighboring agents are defined by the dashed outline rectangles.

**Cooperative predation (C. Predation)**. We introduce a novel continuous domain task involving multiple predator agents aiming to capture a single prey. This environment presents a cooperate-versus-defect dilemma. All predators cooperating (approaching the prey) results in each gaining a reward of $-1$. Universal defection (not approaching) leads to a $-3$ reward for each predator. In mixed scenarios, predators actively pursuing the prey receive a $-4$ reward, while non-participating predators gain 0. The challenge is to encourage agents to cooperate to capture the prey rather than act selfishly. For each episode, the prey's position, $x_{tar} \in \mathcal{X}$, and the agents' starting positions, $x_{ag_i} \in \mathcal{X}$, are randomized within $\mathcal{X} = [0, 30]$. The state is represented as $s^t = [x^t_{ag_1} - x_{tar}, \ldots, x^t_{ag_N} - x_{tar}]$, a continuous variable. The action set $\mathcal{A} = \{-1, +1\}$ represents left and right movements. Neighboring agents are those within a normalized distance of 0.1. Fig.8(c) in AppendixA.2 illustrates this environment. The time horizon of an episode is set as 30. Our main experiments use 8 predator agents, while we test with 20 and 30 agents to evaluate the scalability of our algorithm.

### 5.2 Baselines

We consider three baseline algorithms designed to optimize the total return of all agents under individual rewards, providing a fair comparison with our AS framework to demonstrate its competitiveness, despite not relying on value or policy sharing. While many other MARL algorithms are commonly used as baselines in literature, we exclude them from our experiments due to fundamental differences in problem settings.

**Value function parameter sharing (VPS)** (Zhang et al., 2018b) employs a consensus approach to update individual value functions. Each value function update utilizes the agent's unique reward and incorporates a weighted aggregation of value function parameters from neighboring agents.

**Value sharing (VS)** (Du et al., 2022). Each agent independently learns a value function and shares the output values with neighbors. The individual policy network is updated based on the average value.

**Policy parameter sharing (PS)** (Zhang & Zavlanos, 2019) uses consensus updates to learn global policies. Each agent learns a global policy for all agents and then aggregates policy parameters among neighbours. Value functions are learned independently without consensus updates.

All baseline algorithms and our AS algorithm are implemented on the foundation of the same PPO-based MARL algorithm. This ensures that any performance differences stem from the information sharing mechanisms rather than underlying algorithm variations. The hyperparamters used in the algorithms are provided in Appendix A.3. We chose the hyperparameter values based on common practices in the field. For instance, we set the discount factor to 0.99 and used the same clipping threshold as in the original PPO paper (Schulman et al., 2015). Network sizes were determined based on the state and action dimensions of each environment.

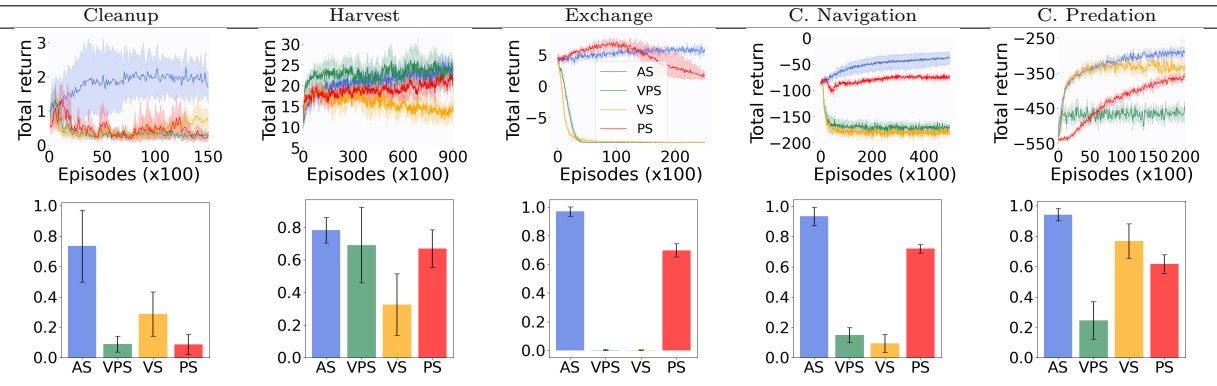

Figure 2: Training curves and normalized final total returns.

## 5.3 Results

We conducted 5 runs with different seeds for each algorithm and environment. Figure 2 shows the training curves and final total returns for different algorithms. Our AS algorithm demonstrates consistent strong performance across all tasks, with total returns matching or exceeding those of baseline algorithms that rely on sharing values or policy parameters. This demonstrates the effectiveness of AS.

**Effect on solving social dilemmas.** AS is specifically designed to handle situations where agents have conflicting individual interests that can hinder collective cooperation, such as in Social Dilemmas. The Exchange task, an extension of the sequential Prisoner's Dilemma, clearly demonstrates the effectiveness of AS in managing these conflicting interests. In the Exchange task, the selfish policy is for each agent to defect (be a free rider) and not open boxes for other agents. However, the collectively optimal solution requires each agent to cooperate and open boxes. Given the reward structure (300 maximum time steps, 0.01 reward loss per time step, 5 reward cost for opening a box, and 10 reward gain for the intended recipient), we can calculate the theoretical optimal returns. The optimal cooperative policy yields approximately 6, while the non-cooperative policy results in -9.

As shown in Figure 2, AS converges to the optimal cooperative policies, achieving a total return close to the theoretical optimum. In contrast, VS and VPS fail to promote cooperation and converge to non-cooperative policies, resulting in lower total returns. This demonstrates AS's ability to encourage cooperation and overcome the challenges posed by the Prisoner's Dilemma in a sequential setting, aligning agents' actions towards the collective goal despite individual incentives for defection.

**Scalability study.** We examine the scalability of our method as the number of agents increases. To reduce communication and computational costs, we implement a sparse network topology and low communication frequency. Our study employs two protocols: (1) each agent randomly selects only one agent for information exchange, and (2) agents communicate every two learning updates (episodes), effectively reducing communication by 50%. During communication gaps, agents do not update anticipated policies of others or use constraints on discrepancies between anticipated and true action distributions. Specifically, we remove the last two terms in Eq. 16 and the policy ratio $\xi_{\mathcal{N}_i}$ involving others' true policies. Consequently, each agent updates its policy independently during these periods. We apply the same approach to baseline algorithms. Figure 3 presents results for C. Predation with 20 and 30 agents, demonstrating our AS algorithm's effectiveness at scale. VS and VPS yield suboptimal results, while PS fails to learn within the given training episodes

**Sensitivity to penalty weight.** We investigate our algorithm's sensitivity to the weight ($\rho$) of the penalty terms. We conduct this analysis across three environments. In Cleanup, we test weights $\rho = 900, 1000$ (used in main experiments), and 1100. For Harvest, we use weights $\rho = 0.01, 0.1$ (used in main experiments), and 1. In C. Predation, we employ weights $\rho = 0.03, 0.1$ (used in main experiments), and 0.3. The range of the weight is related to the value of an appropriate value of the weight, which might be related to the specific action space, such as the dimension of the action space or the meaning of different action dimensions. Figure 4 presents the results of these experiments. The training curves demonstrate that our algorithm

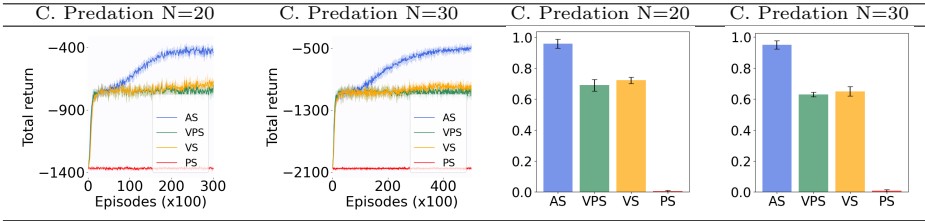

Figure 3: Training curves and normalized final total returns.

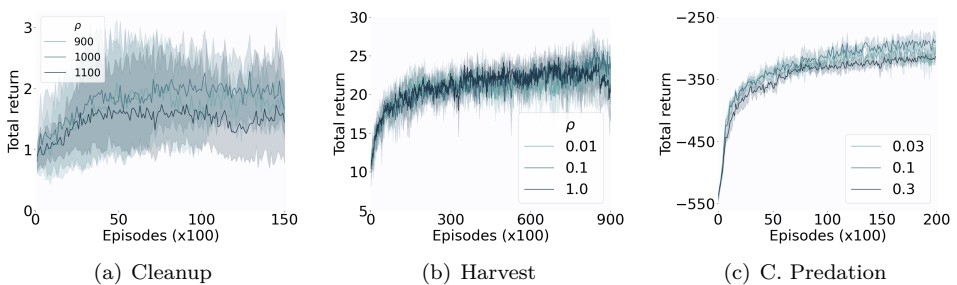

(a) Cleanup

(b) Harvest

(c) C. Predation

Figure 4: Training curves with different penalty weights.

maintains robust performance across a wide range of penalty weights. This robustness suggests that the algorithm's performance is not overly sensitive to precise tuning of this hyperparameter, which is a desirable characteristic for practical applications.

**Ablation study.** We conducted the ablation study with removing the constraints of the objective function, i.e., $\rho = 0$. The experimental results are shown in Figure 5. The results demonstrate that when the constraints are removed, the algorithm performance drops or the agents cannot even learn anything.

**Comparison with Independent learning with individual rewards and Centralized learning with team rewards.** We compared AS with these two additional baselines, which is used in different settings. Specifically, we implemented Independent PPO with individual rewards and Centralized PPO with shared rewards. For the Independent PPO with individual rewards, each agent learns its own policy using single-agent PPO. For the Centralized PPO, we used a single policy network to output probabilities of the joint action distribution and a critic network to evaluate the total return using the total reward. Due to the exponential growth of the joint action space, we limited this experiment to N=2 agents. The results are shown in Figure 6, which indicate that Independent PPO achieves the lowest total return, while Centralized PPO performs the best. Our AS algorithm's performance is closest to that of Centralized PPO, demonstrating its effectiveness in balancing individual privacy and collective performance.

**Anticipated policy and policy discrepancy.** We conduct experiments to investigate the learned anticipated policies and the discrepancy between an agent's policy and the anticipated policies given by another agent. For ease of understanding, we use the task of Cooperative Predation with two agents. For this task, the action set includes two actions: "moving towards the target" and "moving away from the target". The optimal policy that can maximize the collective total returns is both agent moving towards the target. In order to know the anticipated actions learned by each agent, we calculate the proportion of the anticipated actions being "moving towards the target". The results are shown in the top row of Figure 7. The results indicate that both agents anticipate that the other agent can move towards the target rather than move away from the target with a proportion approaching 1. The mean square error between the probability of the action chosen by an agent and the anticipated action given by the other agent is shown in the second row of Figure 7. As the training proceeds, the MSE becomes smaller.

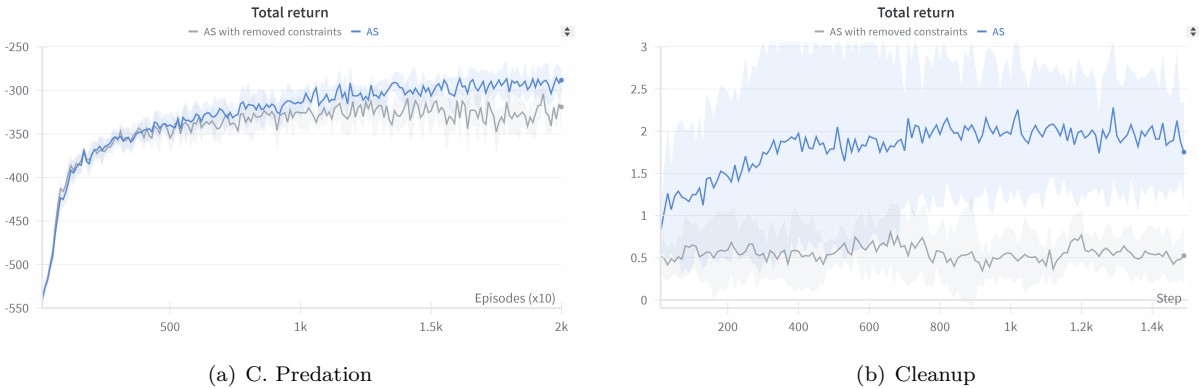

(a) C. Predation

(b) Cleanup

Figure 5: Ablation study of removing constraints.

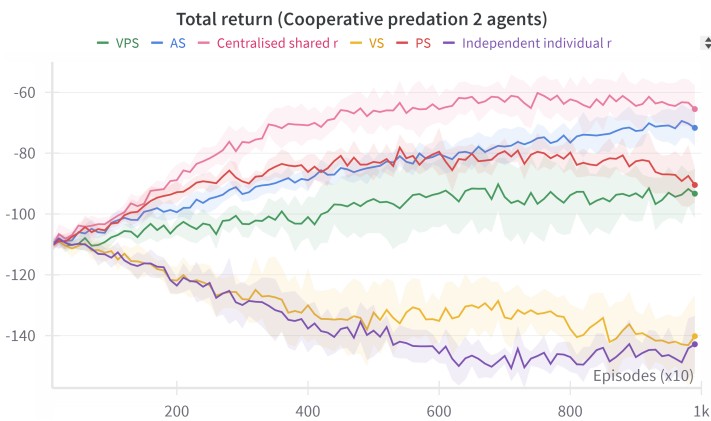

Figure 6: Comparison with Independent learning with individual rewards and Centralized learning with team rewards.

## 6    Discussion and conclusions

In this work, we addressed the challenges of multi-agent cooperation under individual reward conditions, where individual interests may conflict with collective objectives. We introduced Anticipation Sharing (AS) as a solution for scenarios where agents are unaware of others' rewards and policies, and traditional methods of sharing rewards, values, and policy models are infeasible. AS allows agents to incorporate their individual interests into anticipations about other agents' action distributions. Through exchanging these anticipations, agents implicitly build inferences about collective interests, despite the privacy of individual rewards, values, and policies.

Theoretically, we established that the difference between agents' true action distributions and the anticipations from others bounds the discrepancy between individual and collective objectives. This insight led to a novel optimization problem decomposable into individual agents' goals, serving as a lower bound for the original collective objective. Iteratively solving these individual problems drives agents toward cooperative behaviors. Our empirical experiments demonstrate that our algorithm is competitive with the baseline algorithms requiring value or policy parameter sharing.

Our study was primarily motivated by the need to develop a method for cooperative multi-agent learning in settings where agents have individual, private rewards. However, in the course of our experiments, we unexpectedly observed certain limitations of the baseline methods. VS and VPS show inconsistent performance

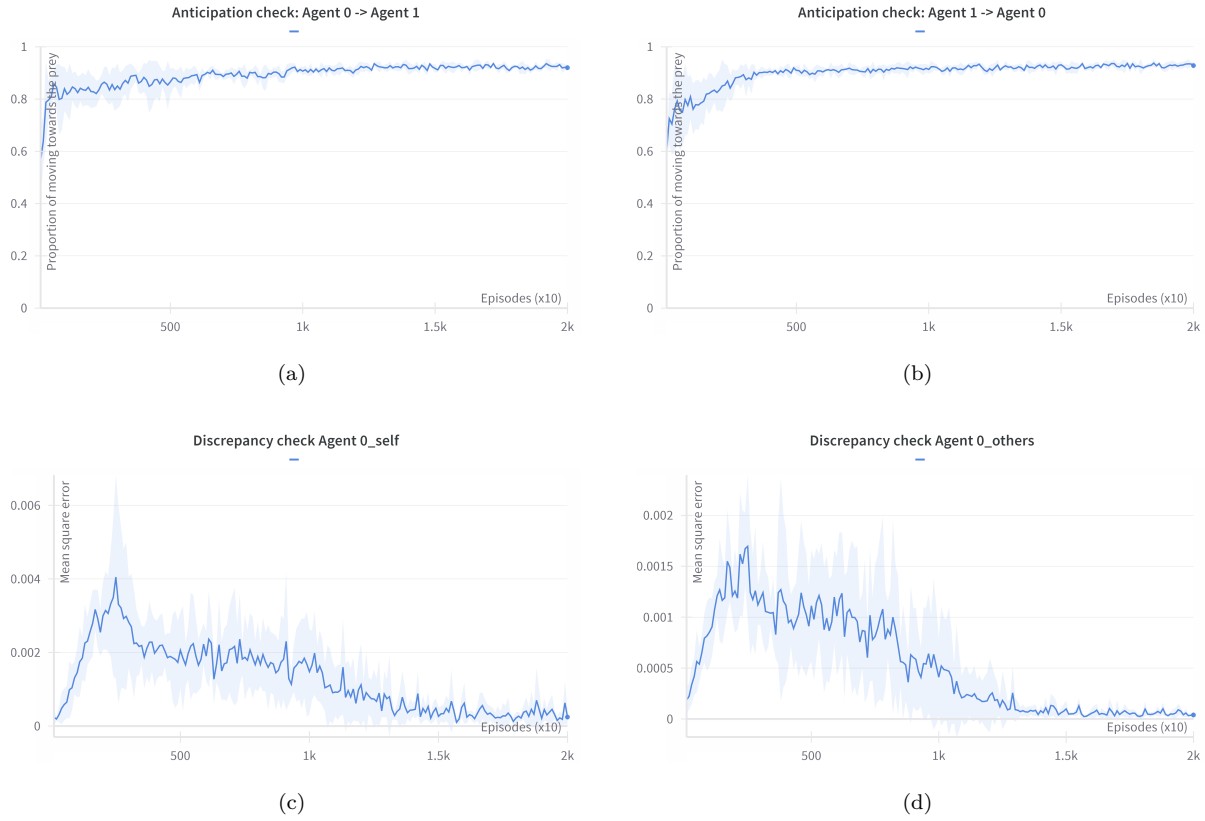

Figure 7: Anticipated action and discrepancy results.

across tasks, performing better in C. Predation compared to Exchange and C. Navigation. This discrepancy likely stems from the varying coordination requirements of these tasks, with Exchange and C. Navigation demanding higher levels of coordination, particularly given the heterogeneous nature of agents with unique individual objectives. VS and VPS, which rely solely on sharing values or value functions to achieve consensus on a system-wide value, may falter in these more complex environments, suggesting that value consensus alone may be insufficient for fostering truly cooperative policies in scenarios requiring intricate coordination. PS, while focusing on direct policy coordination, exhibits slow convergence on some tasks, possibly due to the overhead of sharing entire policy parameters, which may introduce redundant information not essential for effective coordination. These observations highlight the challenges faced by existing methods in achieving effective coordination and computational efficiency. Notably, unlike our AS method, these baseline approaches do not address privacy concerns, as they rely on sharing various forms of information among agents. Our proposed AS method, designed primarily for facilitating cooperation without explicitly sharing rewards, values, or policies, appears to address these challenges effectively while also maintaining the privacy of individual rewards, values and policies. It exchanges action distributions instead of full policy parameters and selectively shares anticipations only with corresponding agents, not all neighbors.

Our work represents an initial step in addressing multi-agent cooperation under private rewards. However, in this work, AS avoids explicit exposure of rewards rather than providing formal privacy guarantees, which qualitatively reduces information sharing compared to methods that directly share rewards or full policies but does not quantitatively minimize the information leakage. In the future work, we will explore techniques to provide stronger privacy guarantees and investigate the trade-off between privacy preservation and cooperative performance. Besides, further reducing algorithm complexity is another promising future direction. Currently, AS algorithm trains $N^2$ policy networks with each agent learning its own policy and anticipated policies about other agents. However, we believe there are solutions to reduce the complexity. For example, we can

use a more computationally efficient network structure, such as using a multi-head policy network that has $N$ output units to output the agent's own policy and $N - 1$ anticipated policies.

Other future research directions include analyzing the convergence properties of our algorithm, applying AS to more complex tasks, refining individual objectives through tighter bounds on discrepancies between individual and collective interests, and exploring alternative optimization strategies based on the AS framework. Additionally, investigating the integration of communication mechanisms from physical and network layers into AS presents another promising avenue for research.

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

# A  Appendix

## A.1  Proofs

### A.1.1  Proof of Lemma 1

**Lemma 1** The following bound holds for the difference between the expected returns of the current policy $\boldsymbol{\pi}_{old}$ and another policy $\boldsymbol{\pi}_{new}$

$$\eta(\boldsymbol{\pi}_{new}) \geq \eta(\boldsymbol{\pi}_{old}) + \zeta_{\boldsymbol{\pi}_{old}}(\boldsymbol{\pi}_{new}) - C \cdot D_{KL}^{max}(\boldsymbol{\pi}_{old}||\boldsymbol{\pi}_{new}), \tag{19}$$

where

$$
\begin{aligned}
\zeta_{\boldsymbol{\pi}_{old}}(\boldsymbol{\pi}_{new}) &= \mathbb{E}_{s\sim d^{\boldsymbol{\pi}_{old}}(s), \boldsymbol{a}\sim\boldsymbol{\pi}_{new}(\cdot|s)} \left[ \sum_i A_i^{\boldsymbol{\pi}_{old}}(s, \boldsymbol{a}) \right], \\
C &= \frac{4 \max_{s,\boldsymbol{a}} |\sum_i A_i^{\boldsymbol{\pi}_{old}}(s, \boldsymbol{a})|\gamma}{(1-\gamma)^2} \\
D_{KL}^{max}(\boldsymbol{\pi}_{old}||\boldsymbol{\pi}_{new}) &= \max_s D_{KL}(\boldsymbol{\pi}_{old}(\cdot|s)||\boldsymbol{\pi}_{new}(\cdot|s)).
\end{aligned}
\tag{20}
$$

**Lemma 3** *Given two joint policies $\boldsymbol{\pi}_{old}$ and $\boldsymbol{\pi}_{new}$,*

$$\eta(\boldsymbol{\pi}_{new}) = \eta(\boldsymbol{\pi}_{old}) + \mathbb{E}_{\tau\sim\boldsymbol{\pi}_{new}} \left[ \sum_{i=1}^{N} \sum_{t=0}^{\infty} \gamma^t A_i^{\boldsymbol{\pi}_{old}}(s_t, \boldsymbol{a}_t) \right], \tag{21}$$

where $\mathbb{E}_{\tau\sim\boldsymbol{\pi}_{new}}[\cdot]$ means the expectation is computed over trajectories where the initial state distribution $s_0 \sim d(s_0)$, action selection $\boldsymbol{a}_t \sim \boldsymbol{\pi}_{new}(\cdot|s_t)$, and state transitions $s_{t+1} \sim \mathcal{P}(\cdot|s_t, \boldsymbol{a}_t)$.

*Proof:* The expected discounted reward of the joint policy, i.e., Eq. 1, can be expressed as

$$\eta(\boldsymbol{\pi}) = \sum_{i=1}^{N} \mathbb{E}_{s_0\sim d(s_0)} \left[ V_i^{\boldsymbol{\pi}}(s_0) \right]. \tag{22}$$

Using $A_i^{\boldsymbol{\pi}_{old}}(s_t, \boldsymbol{a}_t) = \mathbb{E}_{s'}[r_t^i + \gamma V_i^{\boldsymbol{\pi}_{old}}(s') - V_i^{\boldsymbol{\pi}_{old}}(s)]$, we have

$$
\begin{aligned}
&\mathbb{E}_{\tau \sim \boldsymbol{\pi}_{new}} \left[ \sum_{i=1}^{N} \sum_{t=0}^{\infty} \gamma^t A_i^{\boldsymbol{\pi}_{old}}(s_t, \boldsymbol{a}_t) \right] \\
&= \mathbb{E}_{\tau \sim \boldsymbol{\pi}_{new}} \left[ \sum_{i=1}^{N} \sum_{t=0}^{\infty} \gamma^t (r_t^i + \gamma V_i^{\boldsymbol{\pi}_{old}}(s_{t+1}) - V_i^{\boldsymbol{\pi}_{old}}(s_t)) \right] \\
&= \mathbb{E}_{\tau \sim \boldsymbol{\pi}_{new}} \left[ \sum_{i=1}^{N} \sum_{t=0}^{\infty} \gamma^{t+1} V_i^{\boldsymbol{\pi}_{old}}(s_{t+1}) - \sum_{t=0}^{\infty} \gamma^t V_i^{\boldsymbol{\pi}_{old}}(s_t) + \sum_{t=0}^{\infty} \gamma^t r_t^i \right] \\
&= \mathbb{E}_{\tau \sim \boldsymbol{\pi}_{new}} \left[ \sum_{i=1}^{N} \sum_{\mathbf{t=1}}^{\infty} \gamma^t V_i^{\boldsymbol{\pi}_{old}}(s_t) - \sum_{t=0}^{\infty} \gamma^t V_i^{\boldsymbol{\pi}_{old}}(s_t) + \sum_{t=0}^{\infty} \gamma^t r_t^i \right] \\
&= \mathbb{E}_{\tau \sim \boldsymbol{\pi}_{new}} \left[ \sum_{i=1}^{N} (-V_i^{\boldsymbol{\pi}_{old}}(s_0) + \sum_{t=0}^{\infty} \gamma^t r_t^i) \right] \\
&= -\sum_{i=1}^{N} \mathbb{E}_{s_0 \sim d(s_0)}[V_i^{\boldsymbol{\pi}_{old}}(s_0)] + \sum_{i=1}^{N} \mathbb{E}_{\tau \sim \boldsymbol{\pi}_{new}} \left[ \sum_{t=0}^{\infty} \gamma^t r_t^i \right] \\
&= -\eta(\boldsymbol{\pi}_{old}) + \eta(\boldsymbol{\pi}_{new}).
\end{aligned}
\tag{23}
$$

Thus, we have Eq. 21.

Define an expected joint advantage $\bar{A}_{joint}$ as

$$
\bar{A}_{joint}(s) = \mathbb{E}_{\boldsymbol{a} \sim \boldsymbol{\pi}_{new}(\cdot|s)} \left[ \sum_{i=1}^{N} A_i^{\boldsymbol{\pi}_{old}}(s, \boldsymbol{a}) \right].
\tag{24}
$$

Define $L_{\boldsymbol{\pi}_{old}}(\boldsymbol{\pi}_{new})$ as

$$
\begin{aligned}
L_{\boldsymbol{\pi}_{old}}(\boldsymbol{\pi}_{new}) &= \eta(\boldsymbol{\pi}_{old}) + \mathbb{E}_{\tau \sim \pi_{old}} \left[ \sum_{t=0}^{\infty} \gamma^t \bar{A}_{joint}(s_t) \right] \\
&= \eta(\boldsymbol{\pi}_{old}) + \sum_{s} \sum_{t=0}^{\infty} \gamma^t P(s_t = s | \boldsymbol{\pi}_{old}) \bar{A}_{joint}(s).
\end{aligned}
\tag{25}
$$

Leveraging the Lemma 2, Lemma 3, and Theorem 1 provided by TRPO (Schulman et al., 2015), we have

$$
|\eta(\boldsymbol{\pi}_{new}) - L_{\boldsymbol{\pi}_{old}}(\boldsymbol{\pi}_{new})| \leq C \cdot (\max_{s} D_{TV}(\boldsymbol{\pi}_{old}(\cdot|s)||\boldsymbol{\pi}_{new}(\cdot|s)))^2.
\tag{26}
$$

Based on the relationship: $(D_{TV}(p||q))^2 \leq D_{KL}(q||q)$, we have

$$
|\eta(\boldsymbol{\pi}_{new}) - L_{\boldsymbol{\pi}_{old}}(\boldsymbol{\pi}_{new})| \leq C \cdot D_{KL}^{max}(\boldsymbol{\pi}_{old}||\boldsymbol{\pi}_{new}).
\tag{27}
$$

For the second term of the RHS of Eq. 25, we have the following equivalent form

$$
\begin{aligned}
\sum_s \sum_{t=0}^{\infty} & \gamma^t P(s_t = s | \boldsymbol{\pi}_{old}) \bar{A}_{joint}(s) \\
&= \sum_s \sum_{t=0}^{\infty} \gamma^t P(s_t = s | \boldsymbol{\pi}_{old}) \bar{A}_{joint}(s) \\
&= \sum_s d^{\boldsymbol{\pi}_{old}}(s) \bar{A}_{joint}(s) \\
&= \sum_s d^{\boldsymbol{\pi}_{old}}(s) \mathbb{E}_{\boldsymbol{a} \sim \boldsymbol{\pi}_{new}(\cdot|s)} \left[ \sum_{i=1}^{N} A_i^{\boldsymbol{\pi}_{old}}(s, \boldsymbol{a}) \right] \\
&= \zeta_{\boldsymbol{\pi}_{old}}(\boldsymbol{\pi}_{new}),
\end{aligned}
\tag{28}
$$

where $d^{\boldsymbol{\pi}}$ denotes the state visitation distribution under policy $\boldsymbol{\pi}$, and the third line is derived based on the property $d^{\boldsymbol{\pi}_{old}}(s) = P(s_0 = s) + \gamma P(s_1 = s) + \gamma^2 P(s_2 = s) + \cdots$. Thus, we have $L_{\boldsymbol{\pi}_{old}}(\boldsymbol{\pi}_{new}) = \eta(\boldsymbol{\pi}_{old}) + \zeta_{\boldsymbol{\pi}_{old}}(\boldsymbol{\pi}_{new})$. Then, replacing $L_{\boldsymbol{\pi}_{old}}(\boldsymbol{\pi}_{new})$ in Eq. 27, we have

$$
|\eta(\boldsymbol{\pi}_{new}) - (\eta(\boldsymbol{\pi}_{old}) + \zeta_{\boldsymbol{\pi}_{old}}(\boldsymbol{\pi}_{new}))| \leq C \cdot D_{KL}^{max}(\boldsymbol{\pi}_{old} || \boldsymbol{\pi}_{new}),
\tag{29}
$$

and thus Theorem 1 is proved.

### A.1.2 Proof of Lemma 2

**Lemma 2** *The discrepancy between $\zeta_{\boldsymbol{\pi}'}(\tilde{\boldsymbol{\Pi}})$ and the sum of the expected individual advantages calculated with policy $\boldsymbol{\pi}'$ over the true joint policy $\boldsymbol{\pi}$, i.e., $\zeta_{\boldsymbol{\pi}'}(\boldsymbol{\pi})$, is upper bounded as follows.*

$$
\zeta_{\boldsymbol{\pi}'}(\tilde{\boldsymbol{\Pi}}) - \zeta_{\boldsymbol{\pi}'}(\boldsymbol{\pi}) \leq f^{\boldsymbol{\pi}'} + \sum_i \frac{1}{2} \max_{s,\boldsymbol{a}} \left| A_i^{\boldsymbol{\pi}'}(s, \boldsymbol{a}) \right| \cdot \sum_{s,\boldsymbol{a}} \left( \tilde{\boldsymbol{\pi}}^i(\boldsymbol{a}|s) - \boldsymbol{\pi}(\boldsymbol{a}|s) \right)^2,
\tag{30}
$$

*where*

$$
f^{\boldsymbol{\pi}'} = \sum_i \frac{1}{2} \max_{s,\boldsymbol{a}} \left| A_i^{\boldsymbol{\pi}'}(s, \boldsymbol{a}) \right| \cdot |\mathcal{A}| \cdot \|d^{\boldsymbol{\pi}'}\|_2^2,
\tag{31}
$$

*and $\|d^{\boldsymbol{\pi}'}\|_2^2 = \sum_s (d^{\boldsymbol{\pi}'}(s))^2$.*

*Proof:*

$$
\begin{aligned}
\zeta_{\boldsymbol{\pi}'}(\tilde{\boldsymbol{\Pi}}) - \zeta_{\boldsymbol{\pi}'}(\boldsymbol{\pi}) &= \sum_i \mathbb{E}_{s \sim d^{\boldsymbol{\pi}'}(s), \boldsymbol{a} \sim \tilde{\boldsymbol{\pi}}^i(\boldsymbol{a}|s)} \left[ A_i^{\boldsymbol{\pi}'}(s, \boldsymbol{a}) \right] - \mathbb{E}_{s \sim d^{\boldsymbol{\pi}'}(s), \boldsymbol{a} \sim \boldsymbol{\pi}(\boldsymbol{a}|s)} \left[ A_i^{\boldsymbol{\pi}'}(s, \boldsymbol{a}) \right] \\
&= \sum_i \sum_{s,\boldsymbol{a}} d^{\boldsymbol{\pi}'}(s) (\tilde{\boldsymbol{\pi}}^i(\boldsymbol{a}|s) - \boldsymbol{\pi}(\boldsymbol{a}|s)) A_i^{\boldsymbol{\pi}'}(s, a), \\
&\leq \sum_i \max_{s,\boldsymbol{a}} \left| A_i^{\boldsymbol{\pi}'}(s, \boldsymbol{a}) \right| \cdot \left| \sum_{s,\boldsymbol{a}} d^{\boldsymbol{\pi}'}(s) \left( \tilde{\boldsymbol{\pi}}^i(\boldsymbol{a}|s) - \boldsymbol{\pi}(\boldsymbol{a}|s) \right) \right| \\
&\leq \sum_i \max_{s,\boldsymbol{a}} \left| A_i^{\boldsymbol{\pi}'}(s, \boldsymbol{a}) \right| \cdot \sum_{s,\boldsymbol{a}} \frac{1}{2} \left( d^{\boldsymbol{\pi}'}(s)^2 + \left( \tilde{\boldsymbol{\pi}}^i(\boldsymbol{a}|s) - \boldsymbol{\pi}(\boldsymbol{a}|s) \right)^2 \right) \\
&= \sum_i \frac{1}{2} \max_{s,\boldsymbol{a}} \left| A_i^{\boldsymbol{\pi}'}(s, \boldsymbol{a}) \right| \cdot \sum_{s,\boldsymbol{a}} \left( d^{\boldsymbol{\pi}'}(s)^2 + \left( \tilde{\boldsymbol{\pi}}^i(\boldsymbol{a}|s) - \boldsymbol{\pi}(\boldsymbol{a}|s) \right)^2 \right) \\
&= \sum_i \frac{1}{2} \max_{s,\boldsymbol{a}} \left| A_i^{\boldsymbol{\pi}'}(s, \boldsymbol{a}) \right| \cdot \left( |\mathcal{A}| \cdot \|d^{\boldsymbol{\pi}'}\|_2^2 + \sum_{s,\boldsymbol{a}} \left( \tilde{\boldsymbol{\pi}}^i(\boldsymbol{a}|s) - \boldsymbol{\pi}(\boldsymbol{a}|s) \right)^2 \right) \\
&= f^{\boldsymbol{\pi}'} + \sum_i \frac{1}{2} \max_{s,\boldsymbol{a}} \left| A_i^{\boldsymbol{\pi}'}(s, \boldsymbol{a}) \right| \cdot \sum_{s,\boldsymbol{a}} \left( \tilde{\boldsymbol{\pi}}^i(\boldsymbol{a}|s) - \boldsymbol{\pi}(\boldsymbol{a}|s) \right)^2
\end{aligned}
\tag{32}
$$

where

$$
f^{\boldsymbol{\pi}'} = \sum_i \frac{1}{2} \max_{s,\boldsymbol{a}} \left| A_i^{\boldsymbol{\pi}'}(s, \boldsymbol{a}) \right| \cdot |\mathcal{A}| \cdot \|d^{\boldsymbol{\pi}'}\|_2^2.
$$

### A.1.3 Proof of Theorem 1

**Theorem 1** *The discrepancy between the return of the newer joint policy and the value of $\zeta_{\boldsymbol{\pi}_{old}}(\tilde{\boldsymbol{\Pi}}_{new})$ is lower bounded as follows:*

$$
\begin{aligned}
\eta(\boldsymbol{\pi}_{new}) - \zeta_{\boldsymbol{\pi}_{old}}(\tilde{\boldsymbol{\Pi}}_{new}) \geq{}& \eta(\boldsymbol{\pi}_{old}) - C \cdot \sum_i D_{KL}^{max}(\pi_{old}^{ii} || \pi_{new}^{ii}) - f^{\boldsymbol{\pi}_{old}} \\
& - \sum_i \frac{1}{2} \max_{s,\boldsymbol{a}} |A_i^{\boldsymbol{\pi}_{old}}(s,\boldsymbol{a})| \cdot \sum_{s,\boldsymbol{a}} \left( \tilde{\boldsymbol{\pi}}_{new}^i(\boldsymbol{a}|s) - \boldsymbol{\pi}_{new}(\boldsymbol{a}|s) \right)^2 .
\end{aligned}
\tag{33}
$$

*Proof:* According to Theorem 1, we have

$$
\eta(\boldsymbol{\pi}_{new}) \geq \zeta_{\boldsymbol{\pi}_{old}}(\boldsymbol{\pi}_{new}) + \eta(\boldsymbol{\pi}_{old}) - C \cdot D_{KL}^{max}(\boldsymbol{\pi}_{old} || \boldsymbol{\pi}_{new}).
\tag{34}
$$

The KL divergence has the following property (Su & Lu, 2022):

$$
D_{KL}^{max}(\boldsymbol{\pi}_{old} || \boldsymbol{\pi}_{new}) \leq \sum_i D_{KL}^{max}(\pi_{old}^{ii} || \pi_{new}^{ii}).
\tag{35}
$$

Based on Eq. 34 and Eq. 35, we have

$$
\eta(\boldsymbol{\pi}_{new}) \geq \zeta_{\boldsymbol{\pi}_{old}}(\boldsymbol{\pi}_{new}) + \eta(\boldsymbol{\pi}_{old}) - C \cdot \sum_i D_{KL}^{max}(\pi_{old}^{ii} || \pi_{new}^{ii}).
\tag{36}
$$

Using Theorem 2, $\zeta_{\boldsymbol{\pi}_{old}}(\tilde{\boldsymbol{\Pi}}_{new})$ and $\zeta_{\boldsymbol{\pi}_{old}}(\boldsymbol{\pi}_{new})$ satisfy the following inequality:

$$
\begin{aligned}
& \zeta_{\boldsymbol{\pi}_{old}}(\boldsymbol{\pi}_{new}) \\
& \geq \zeta_{\boldsymbol{\pi}_{old}}(\tilde{\boldsymbol{\Pi}}_{new}) - \sum_i \frac{1}{2} \max_{s,\boldsymbol{a}} |A_i^{\boldsymbol{\pi}_{old}}(s,\boldsymbol{a})| \cdot \sum_{s,\boldsymbol{a}} \max_s d^{\boldsymbol{\pi}_{old}}(s)^2 + (\tilde{\boldsymbol{\pi}}_{new}^i(\boldsymbol{a}|s) - \boldsymbol{\pi}_{new}(\boldsymbol{a}|s))^2 .
\end{aligned}
\tag{37}
$$

According to Eq. 31, Eq. 37 can be transformed as:

$$
\begin{aligned}
& \zeta_{\boldsymbol{\pi}_{old}}(\boldsymbol{\pi}_{new}) \\
& \geq \zeta_{\boldsymbol{\pi}_{old}}(\tilde{\boldsymbol{\Pi}}_{new}) - f^{\boldsymbol{\pi}_{old}} - \sum_i \frac{1}{2} \max_{s,\boldsymbol{a}} |A_i^{\boldsymbol{\pi}_{old}}(s,\boldsymbol{a})| \cdot \sum_{s,\boldsymbol{a}} \left( \tilde{\boldsymbol{\pi}}_{new}^i(\boldsymbol{a}|s) - \boldsymbol{\pi}_{new}(\boldsymbol{a}|s) \right)^2 .
\end{aligned}
\tag{38}
$$

By replacing $\zeta_{\boldsymbol{\pi}_{old}}(\boldsymbol{\pi}_{new})$ in Eq. 36 with the RHS of Eq. 38, we can get Eq. 33, and thus Theorem 1 is proved.

## A.2 Illustrations of simulated environments

We evaluate the performance of our AS algorithm across a spectrum of tasks, spanning both discrete (Exchange and Cooperative Navigation) and continuous (Cooperative Predation) spaces. An illustration of these environments is shown in Fig. 8.

## A.3 Hyperparameters

Hyperparameters used in our experiments are given in Tables 1 and 2.

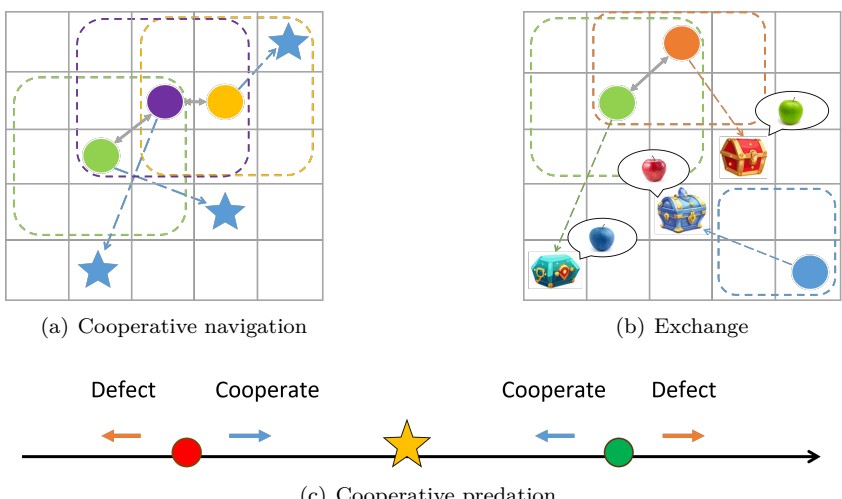

(a) Cooperative navigation         (b) Exchange

(c) Cooperative predation

Figure 8: Illustrations of environments.

Table 1: Common hyperparameters used in all environments.

| Hyperparameter | Value |
|---|---|
| Critic learning rate | 1e-4 |
| Discount factor $\gamma$ | 0.99 |
| GAE $\lambda$ | 0.98 |
| Clipping $\epsilon$ | 0.2 |
| Update iteration $K$ | 3 |
| Activation | ReLU |
| Optimizer | Adam |

Table 2: Hyperparameters used in different environments.

| Domain | Cleanup | Harvest | C. Predation | C. Navigation | Exchange |
|---|---|---|---|---|---|
| Critic network size | (1024, 256, 1) | (1024, 256, 1) | (128, 64, 1) | (128, 64, 1) | (128, 64, 1) |
| Actor network size | (1024, 256, d_a) | (1024, 256, d_a) | (128, 64, d_a) | (128, 64, d_a) | (128, 64, d_a) |
| Actor learning rate | 5e-5 | 5e-5 | 1e-4 | 5e-5 | 5e-5 |
| $\rho$ | 1e3 | 0.1 | 0.1 | 1e4 | 1e4 |

