# OpenReview forum: "Learning to Cooperate under Private Rewards"
_TMLR — Rejected by TMLR_

### Review · Reviewer_ngjK · 2024-08-03

**Summary Of Contributions:**

This work is concerned with multi-agent reinforcement learning (MARL) in a setting in which agents are trained to optimise the total rewards of all agents, which indicates team performance and coordination, but each agent only has access to its own individual rewards. Typical approaches that aim to optimise similar objectives usually share rewards or policy/ value function models across agents, but such sharing might reveal information and violate privacy of agents and, thus, might be undesirable. To address these concerns, a novel MARL algorithm in Anticipation Sharing (AS) is proposed in which each agent learns and communicates "anticipated policies" to all other agents. These anticipated policies reflect the policies an agent would like the other agents to follow in order to maximise its own individual rewards. Agents optimise their policy and value functions by following usual policy gradients and additionally optimising their policies to be close to the anticipated policies of other agents communicated to them using a constrained optimisation process. AS is shown to reach similar or higher evaluation returns than baseline algorithms in several tasks within the setting, all while preserving privacy.

**Audience:**

Yes

**Broader Impact Concerns:**

I do not foresee ethical implications that require further discussion.

**Claims And Evidence:**

No

**Requested Changes:**

1. The work would be substantially strengthened by clearing up the problem setting and well motivating its relevance. As discussed in Weakness 1., the setting is currently rather confusing and not well motivated. I suggest to
	1. Add a thorough discussion of the setting and how it relates and differs from other well-understood settings (in particular individual reward formalisms like stochastic games and common reward formalisms like Dec-POMDPs).
	2. Reconsider the currently presented motivational examples (that I find not very convincing; as discussed in Weakness 1.3), and present few examples in which your setting is relevant, i.e. privacy needs to be preserved and neither common-reward nor individual reward optimisation would be sufficient/ applicable.
2. Correct the mistakes in the proofs and clarify unclear steps within the derivations (see Weakness 2.).
3. How are the anticipated policies of each agent $\pi^{ij}$ trained? How does this training process lead to anticipation policies that reflect agent $i$'s preferences with respect to their individual rewards/ returns for the policies of the other agent $j$? (see Weakness 3.1)
4. Could it be that the communication of anticipated policies to other agents does not preserve privacy but reveals significant information about the preferences of each agent? Further discussion and ideally an experiment would be helpful to clear this up since it is central to the claims about AS and its purpose within the proposed setting. (see Weakness 3.2)
5. Clarify the implementation details of baseline algorithms (how are they similar/ different to AS), and clarify what objectives they are optimising (is it also total returns as defined in Eq. 1)? (see Weakness 4.1)
6. Add two new baselines in which you train standard MARL algorithms with similar implementation details to AS from individual rewards, and purely from shared common rewards (e.g. by defining common rewards as the sum of all individual agents' rewards). (see Weakness 4.1)
7. Clarify how hyperparameters of AS and all baselines were optimised for the experiments. (see Weakness 4.1)
8. Add some analysis that discusses how the anticipated policies and their constraints play out within the experiments. What anticipated policies are agents learning? How similar are these anticipated policies to the policies already learned by other agents? (see Weakness 4.3)
9. Provide insight to the computational cost of AS and baselines across the experiments with few and many agents. (see Weakness 4.5)
10. Provide more insight into the sensitivity study of $\rho$. What happens if you use $\rho = 1e4$ in Harvest and C. Predator? What about $\rho = 0.1$ n Cleanup, C. Navigation, and Exchange? (see Weakness 4.6)
11. Further discussion of related work, outlined in Weakness 5., would be helpful to understand the relation of this work to existing literature.

**Strengths And Weaknesses:**

## Strengths
The general setting of privacy-preserving MARL training is, to the best of my knowledge, a newly considered setting with potential interest in several specific use cases. I also like how the work proposes a new algorithm alongside theory that supports the algorithm design, sheds light on what might be required in this new setting, and demonstrates benefits in empirical experiments. The writing is mostly clear and the paper well structured.

## Weakness

### 1. Unclear / confusing problem setting

1. Has the "multi-agent Markov decision process" (MMDP) been formalised like this before? Do you have any citations/ prior work using this formalism?
	- It appears almost identical to Stochastic Games [1] / Markov Games [2] with the exception that the optimisation objective is for agents to maximise the sum of all agents' returns?
	- Meanwhile the optimisation objective appears similar to Dec-POMDP [3] with all agents optimising a joint reward signal
	- Due to the unusual nature of the setting, a more thorough discussion of the setting, similar problem settings, and the implications of these differences would be valuable to bring readers on the same page
2. "Their individual rewards may conflict, yet they must cooperate to attain a collective optimum" (Section 3, last paragraph) and "Under individual rewards [...] prioritising individual rewards can produce suboptimal collective outcomes." (Section 1)
	- These conflicts arise due to agents optimising their own individual rewards that might conflict with each other. However, in the setting defined in Section 3 all agents optimise for the same optimisation objective (total returns of all agents in Eq. 1), even if they do not have access to the rewards of other agents. Under this shared and common optimisation objective, these conflicts might not arise.
3. Lastly, I am unsure how convincing I find the motivation of this setting in which agents try to optimise a common optimisation objective in the total returns of all agents but only have access to individual rewards. In the introduction you discuss few applications but none of these appear natural under your formalism but instead could be captured by the common settings of optimising individual rewards [1, 2] or sharing and optimising a single common reward across agents:
	- Competitive market places --> companies have no common goal but only try to maximise their own profits --> this would be reflected in individual reward optimisation with each agent optimising their individual returns (see [1, 2]).
	- Multi-robot task allocation --> if all agents have a common goal (as stated in the description), even if agents have individual energy constraints and local sensors, this can be formalised and optimised as a common-reward problem [3]. The energy constraints of all agents could be integrated into the common reward function and local sensors merely imply that the task is partially observable with each agent receiving their own local observations
	- Autonomous delivery --> each delivery robot would try to deliver as fast as possible, arguably without caring about other robots also being fast so this could be realised with each agents optimising their own individual rewards (see [1, 2]). If instead, all robots should optimise for a common solution such that collectively they are as efficient as possible, then this should and could be implemented as a common reward problem without a need for privacy (see [3]).

### 2. Theory contributions

1. I believe there are two mistakes in the proofs provided within Appendix A.
	- In Eq. 23 in the 2nd last line, the total expected returns of the old and new policies are split. However, the expectation of the previous line is over $\tau \sim \pi_{new}$ and now the left expectation is over $s_0$ without explicitly denoting from which distribution this state is sampled from. As far as I can tell, this state should be under the expectation $s_0 \sim d^{\pi_{new}}$ since the trajectories of the previous expectation are sampled from $\pi_{new}$. This however means that the left sum /expression does **not** equal to $\upeta(\pi_{old})$  since that by definition in Eq. 1 would require the expectation to be over $\tau \sim \pi_{old}$ but it is over $\pi_{new}$! I am unsure how easily the following proofs can be salvaged without this derivation in Eq. 23 holding but to me it seems this derivation was essential for following proofs that might now not hold anymore.
	- In Eq. 28 in the third line you write $\bar{A}_{joint}(s_t)$ but $s_t$ is undefined since the sum over $t$ was resolved. I am unsure what you meant to express here and why this step holds.
2. Beyond these mistakes, the following steps within the proofs are unclear to me. Any more details explanations as to why these steps hold would be appreciated:
	- In Eq. 23, it is unclear to me how you go from line 2 to line 3 where you remove the sum over $t$, and pull out $V_i^{\pi_{old}}(s_0)$ and the sum of discounted returns.
	- In Eq. 23, you use $d^{\pi'}(s)$ to denote the visitation probability of state $s$ when following policy $\pi'$ as far as I understand. However, I am unsure what $d^{\pi'}$ is meant to express which appears in the 2nd last line. Could you explain what this represents and where it comes from?

### 3. Anticipation sharing algorithm

1. Within the AS algorithm, each agent $i$ optimises $N$ (number of agents) policies with $\pi^{ii}$ denoting the policy that the agent itself follows and $\pi^{ij}$ denoting the anticipated policy of agent $i$ for agent $j$. I see that each agent optimises the constrained optimisation objective to incentivise them to train their policy $\pi^{ii}$ to stay close to the anticipated policies for them of other agent $\pi^{ji}$ as given in the 2nd term of the 2nd line in Eq. 16. But I don't fully follow how agents optimise their anticipated policies for other agents in $\pi^{ij}$. According to the first term within the 2nd line of Eq. 16, I would think agent $i$ merely tries to optimise $\pi^{ij}$ to be close to $\pi^{jj}$. How does this lead to anticipation policies of other agents that reflect agent $i$'s preferences with respect to their individual rewards/ returns?
2. The proposed algorithm of AS is motivated by the setting in which all agents want to optimise a single objective of total agent returns (Eq. 1) but have only access to their own individual rewards since sharing a common reward would not preserve privacy about agents' goals. However, it is unclear to me whether agents sharing anticipated policies, as proposed in AS, with other agents itself preserves privacy. Agents communicate these anticipated policies to other agents that are meant to reflect an agent's preferences over other agents' policies (see question 1. above about how this is achieved) but such preferences in themselves might reveal a lot about an agent's rewards. For example, if agent $i$ states to prefer agent $j$ to apply a certain action $a_j$ in state $s$ over other agents in their anticipated policy, that indicates that the reward function of agent $i$, $R_i$, assigns higher reward to the resulting transition or to future states compared to transitions with other actions. Any discussion as to how much privacy is preserved by AS would be appreciated.
	- To investigate how much "leakage" of information about the reward of other agents there is, one could design an experiment in which each agent trains a model that tries to predict the rewards of other agents based on such preference information (if agent $i$'s anticipated policy for agent $j$'s prefers action $a_j$ over other actions in a particular state, increase the "preference/ reward" of this action over other agents). If such a model can be trained to accurately represent the preferences of agent $i$ over possible outcomes/ actions, then significant correlation between rewards and communicated anticipated policies exists and privacy would not be preserved through this sharing of information.

### 4. Experiments

1. **Baselines:**
	- It appears that the AS algorithm is built on top of/ similar to independent PPO (IPPO) which has been shown to be a strong baseline in MARL (e.g. [4, 5]). Were the baselines (sharing value functions, value estimates, policy parameters) also implemented on the foundation of the same MARL algorithm with similar implementation details (e.g. use of PPO surrogate objective, GAE returns, ...)? Otherwise, results between algorithms might differ based on the underlying MARL algorithm rather than the process of sharing information.
	- AS is designed to optimise the sum of total rewards of all agents (Eq. 1). Are the baselines optimising for the same objective?
	- I would expect two additional simple baselines to better understand the validity of the setting and chosen approach. Both should be implemented on top of a MARL algorithm with similar implementation details to AS (e.g. IPPO as discussed above).
		1. Individual rewards: optimise agents under the typical stochastic game/ Markov game formalism with each agent receiving individual rewards and optimising their individual returns. This approach is simple, well understood, and perfectly preserves privacy (without parameter sharing) so it should be considered.
		2. Shared rewards: in Section 2 you already discuss the setting of all agents sharing and optimising returns over a common / shared reward. This setting [3] is well understood and simple to optimise. It might not preserve privacy but optimises the same objective as AS where all agents try to maximise the total returns across all agents, so it would serve as a helpful "oracle"/ upper bound on the expected performance of AS.
	- How were hyperparameters of AS and the baseline algorithms chosen in this work?

2. **Training curves**:
	- In the Exchange task, the total returns of PS appears to degrade in later stages of training. Why is that?

3. **Training metrics:** The paper only reports learning curves and final performance according to total returns across agents. These are important metrics but do not reveal any insight into the mechanisms of AS. I would expect at least some analysis that shows how well agents are upholding constraints defined by the anticipated policies of other agents throughout training. What anticipated policies are other agents learning? How similar are these to the policies already learned by other agents?

4. **Ablation:** I would expect ablations where the constrainted optimisation components of AS are removed ($\kappa = \rho = 0$) to have this comparison point of an algorithm with identical implementation details to AS but no anticipated policies/ constrained optimisation procedure.

5. **Scalability study:** The work discusses the scalability of different approaches in tasks with larger number of agents. An important consideration so far not reported of this would be to report the computational cost of AS and all baselines. For AS, you train $N^2$ policies with $N$ being the number of agents which might be prohibitively expensive compared to other approaches.

6. **Sensitivity analysis:** In the conducted sensitivity study, it is stated that "[AS] maintains robust performance across a wide range of penalty weights" but the weights appear to be chosen around vastly different values depending on the task. Following Figure 4 and Table 2 in the Appendix, Cleanup, C. Navigation, and Exchange require large values of $\rho$ of ~1e4 while other tasks require small values of ~0.1. I would not consider this to establish robustness. What happens if you use $\rho = 1e4$ in Harvest and C. Predator? What about $\rho = 0.1$ n Cleanup, C. Navigation, and Exchange?

### 5. Related work
1. Due to the similarity of the proposed constrained optimization problem and trust region optimization in the multi-agent context, it appears that [6] is significantly related. This work studies trust region bounds in MARL under decentralized PPO. Any discussion on the similarity of the proposed approach and the trust region formulation in [6] would be helpful.
2. By motivation, the proposed setting appears similar to no-conflict games in which agents have individual rewards but agree on the optimal joint policy. There have been approaches proposed for this setting that leverage this property by optimising agents that assume that other agents will aim to maximise their returns (e.g. [7, 8]). Further discussion of similarity of these settings and potential experiments of these algorithms in your evaluation would be appreciated.

---

### References

[1] Shapley, Lloyd S. (1953). "Stochastic games". PNAS, 39 (10): 1095–1100.

[2] Littman, Michael L. (1994). "Markov games as a framework for multi-agent reinforcement learning". International Conference on Machine Learning.

[3] Bernstein, Daniel S., Robert Givan, Neil Immerman, and Shlomo Zilberstein. "The complexity of decentralized control of Markov decision processes." _Mathematics of operations research_ 27, no. 4 (2002): 819-840.

[4] Papoudakis, Georgios, Filippos Christianos, Lukas Schäfer, and Stefano V. Albrecht. "Benchmarking multi-agent deep reinforcement learning algorithms in cooperative tasks." In _Conference on Neural Information Processing Systems, Track on Datasets and Benchmarks_, 2021.

[5] De Witt, Christian Schroeder, Tarun Gupta, Denys Makoviichuk, Viktor Makoviychuk, Philip HS Torr, Mingfei Sun, and Shimon Whiteson. "Is independent learning all you need in the starcraft multi-agent challenge?." In _arXiv preprint arXiv:2011.09533_, 2020.

[6] Sun, Mingfei, Sam Devlin, Jacob Beck, Katja Hofmann, and Shimon Whiteson. "Trust region bounds for decentralized ppo under non-stationarity." In _International Conference on Autonomous Agents and Multiagent Systems_, 2023.

[7] Littman, Michael L. "Friend-or-foe Q-learning in general-sum games." In _ICML_, vol. 1, no. 2001, pp. 322-328. 2001.

[8] Christianos, Filippos, Georgios Papoudakis, and Stefano V. Albrecht. "Pareto Actor-Critic for Equilibrium Selection in Multi-Agent Reinforcement Learning." In _Transactions on Machine Learning Research Journal_, 2023.

---

> ### Author Response · Authors · 2024-08-25
> **Author Response to Reviewer ngjK**
>
> **Unclear / confusing problem setting**
>
> **1. Has the "multi-agent Markov decision process" (MMDP) been formalised like this before? Do you have any citations/ prior work using this formalism?
> It appears almost identical to Stochastic Games [1] / Markov Games [2] with the exception that the optimisation objective is for agents to maximise the sum of all agents' returns?
> Meanwhile the optimisation objective appears similar to Dec-POMDP [3] with all agents optimising a joint reward signal. Due to the unusual nature of the setting, a more thorough discussion of the setting, similar problem settings, and the implications of these differences would be valuable to bring readers on the same page.**
>
> We appreciate your insightful comments regarding our problem formulation. Our formalization of the multi-agent Markov decision process (MMDP) does share similarities with Stochastic Games/ Markov Games, with the key distinction being our optimization objective of maximizing the sum of all agents' returns. Below please find our response to your points:
>
> (1) *Prior use of this formalism:* The MMDP formulation we use has indeed been employed in previous works. Zhao et al. (2020) and Krouka et al. (2022) formalized the same problem as we did. Chen et al. (2022) considered a similar problem, but with a central controller that can collect information from all agents. Zhang et al. (2018), Du et al. (2022), and Sha et al. (2021) used the same basic problem formalism, but added a network structure on agent systems, referring to it as Networked MMDP or MARL over networks. Additionally, Lei et al. (2022) presented the Networked MARL problem from the perspective of Alternating Direction Method of Multipliers (ADMM).
>
> (2) *Similarities to other frameworks:* We acknowledge the similarity of our formulation to Stochastic Games (Markov Games) in terms of structure, and to Dec-POMDP in terms of the optimization objective. However, there are key distinctions. Unlike standard Stochastic Games, our agents are cooperative and aim to maximize a collective return. Unlike Dec-POMDP, our agents have access to the full state (not partial observations) and individual reward functions (not a shared reward signal).
>
> (3) *Implications of differences:* The main implication of our formulation is that it allows us to study scenarios where agents need to cooperate to maximize collective return, but have access only to their individual rewards. This setting is particularly relevant for applications where privacy concerns or decentralized control are important. It enables us to explore the balance between cooperative behavior and individual privacy in multi-agent systems, which is crucial in many real-world applications.
>
> We have added a more thorough discussion of our problem setting and its relation to similar frameworks in Section 3 accordingly. We believe that these additions will help readers better understand the context and significance of our work.

---

> > ### Author Response · Authors · 2024-08-25
> > **Author Response to Reviewer ngjK**
> >
> > **2. "Their individual rewards may conflict, yet they must cooperate to attain a collective optimum" (Section 3, last paragraph) and "Under individual rewards [...] prioritising individual rewards can produce suboptimal collective outcomes." (Section 1)
> > These conflicts arise due to agents optimising their own individual rewards that might conflict with each other. However, in the setting defined in Section 3 all agents optimise for the same optimisation objective (total returns of all agents in Eq. 1), even if they do not have access to the rewards of other agents. Under this shared and common optimisation objective, these conflicts might not arise.**
> >
> > We appreciate the reviewer's astute observation regarding the apparent inconsistency between our statements about conflicting individual rewards and the formal problem setting where all agents optimize the same objective. This point highlights a need for greater clarity in our presentation.
> >
> > Our problem setting indeed defines a collective objective (Eq. 1) that all agents should ultimately optimize. However, this collective objective is composed of individual reward functions that may conflict. The crux of our problem lies in how agents, with access only to their individual rewards, can learn to optimize this collective objective without explicitly sharing their rewards.
> >
> > When we state that ``individual rewards may conflict,'' we're referring to the potential misalignment between optimizing one's own reward function and optimizing the collective return. An agent na\"{\i}vely optimizing its individual reward might take actions that are suboptimal for the group. For instance, in a traffic routing scenario, an agent might choose a route that's individually optimal but contributes to overall congestion, reducing the collective return. In fact, Social Dilemmas are typical examples, such as Prisoner's Dilemma - although mutual cooperation yields the best collective outcome, rational individuals are tempted to defect, leading to a worse outcome for both.
> >
> > Our AS method bridges this gap between individual reward optimization and collective return maximization. It enables agents to approximate the optimization of the collective objective while operating solely with their individual reward signals. Through the sharing of anticipated actions, agents can implicitly account for the impact of their actions on others, aligning their behavior more closely with the collective optimum.
> >
> > In our formal problem setting, we present the desired end state where agents optimize the collective return. Our method then provides a way to achieve this state starting from a point where agents only have access to individual rewards.
> > In our revised paper, we clarified more explicitly the transition from individual rewards to collective optimization in the last paragraph of Section 3.

---

> ### Author Response · Authors · 2024-08-25
> **Author Response to Reviewer ngjK**
>
> **3. Lastly, I am unsure how convincing I find the motivation of this setting in which agents try to optimise a common optimisation objective in the total returns of all agents but only have access to individual rewards. In the introduction, you discuss a few applications, but none of these appear natural under your formalism but instead could be captured by the common settings of optimising individual rewards [1, 2] or sharing and optimising a single common reward across agents.**
>
> We sincerely appreciate the reviewer's thoughtful critique of our problem motivation. Your comments have highlighted a need for us to articulate more clearly the relevance and uniqueness of our problem setting. We agree that our initial examples could be interpreted as better suited to either individual reward optimization or shared reward scenarios. Allow us to clarify and provide more appropriate motivating examples.
>
> Our work addresses a critical gap in multi-agent reinforcement learning: scenarios where agents must cooperate to achieve a collective goal but where privacy concerns or structural constraints prevent the sharing of individual reward functions.
> Consider the following examples that more accurately reflect our problem setting:
>
> (1) *Supply Chain Optimization:* Multiple companies in a supply chain need to coordinate their production and logistics to maximize overall efficiency. Each company has its own profit function (individual reward) that it wants to keep private due to competitive concerns. However, the overall supply chain performance (collective return) depends on their coordinated actions. Our method allows these companies to cooperate effectively without revealing their individual profit functions.
>
> (2) *Distributed Energy Systems:* In a smart grid, various energy producers and consumers need to coordinate to balance supply and demand. Each entity has its own cost/utility function (individual reward) that it may not want to disclose. Yet, the stability and efficiency of the entire grid (collective return) depend on their coordinated actions. Our approach enables this coordination while preserving the privacy of individual cost/utility functions.
>
> (3) *Federated Learning in Healthcare:* Multiple hospitals aim to collaboratively train a medical diagnosis model without sharing patient data. Each hospital has its own performance metric (individual reward) based on its specific patient population and priorities. The goal is to create a model that performs well across all hospitals (collective return) without compromising individual hospital data or metrics.
>
>
> In these scenarios, unlike in purely individual reward settings, agents have an incentive to cooperate for better collective outcomes. Unlike in shared reward settings, agents have distinct individual reward functions that they cannot or do not want to share explicitly.
> Our AS approach addresses precisely this gap: it enables cooperation towards a collective goal while preserving the privacy of individual reward functions. This approach is particularly relevant in today's privacy-conscious and decentralized operational environments.
>
> We acknowledge that our initial presentation may have not effectively conveyed the unique aspects of our problem setting. In our revised paper, we introduced more appropriate examples that more clearly demonstrate the need for both cooperation and reward privacy in the Introduction and provided a more detailed discussion of how our problem setting differs from both individual reward optimization and shared reward scenarios in Section 3.
>
> **Theory contributions**
>
> **1. I believe there are two mistakes (in Eq. 23 and Eq. 28) in the proofs provided within Appendix A.**
>
> Thanks for pointing them out.
> As for Eq. 23, we believe it is correct. Actually, the state $s_0$ is the initial state of the trajectory, which is sampled from a distribution independent of any policy. But we apologize that we miswrote the denotation of the initial state distribution $s_0 \sim d^{\pi_{new}}(s)$. We have modified it as $s_0 \sim d(s_0)$ in our revised paper.
>
> Thanks for pointing out Eq. 28, it was our oversight that $P(s_t=s|\pi_{old})$ should be included in the summation over $t$. We have corrected it in our revision. Please see Eq. 25 and Eq. 28 in our revised paper.
>
> **2. Beyond these mistakes, the following steps (in Eq. 23 and Eq. 32) within the proofs are unclear to me. Any more details explanations as to why these steps hold would be appreciated.**
>
> Thank you for your questions. We added more steps between line 2 and line 3 in Eq. 23 in our revised paper. We believe it is much clearer now.
>
> In Eq. 32, we define $\\|d^{\pi'}\\|_2^2 = \sum_s (d^{\pi'}(s))^2$, where by abuse of notation, we deem $d^{\pi'}$ as a vector with each $s$ being one dimension of the vector. We added the definition in our revised paper.

---

> > ### Author Response · Authors · 2024-08-25
> > **Author Response to Reviewer ngjK**
> >
> > **Anticipation sharing algorithm**
> >
> > **1. Within the AS algorithm, each agent $i$ optimises $N$ (number of agents) policies with $\pi^{ii}$ denoting the policy that the agent itself follows and $\pi^{ij}$ denoting the anticipated policy of agent $i$ for agent $j$. I see that each agent optimises the constrained optimisation objective to incentivise them to train their policy $\pi^{ii}$ to stay close to the anticipated policies for them of other agent $\pi^{ji}$ as given in the 2nd term of the 2nd line in Eq. 16. But I don't fully follow how agents optimise their anticipated policies for other agents in $\pi^{ij}$. According to the first term within the 2nd line of Eq. 16, I would think agent $i$ merely tries to optimise $\pi^{ij}$ to be close to $\pi^{jj}$. How does this lead to anticipation policies of other agents that reflect agent i's preferences with respect to their individual rewards/ returns?**
> >
> > We appreciate the reviewer's insightful questions about our AS algorithm. Let us clarify the key aspects of how agents optimize their anticipated policies and how these reflect individual preferences. The core of our approach lies in the optimization problem defined in Eq. 12 subject to the constraints (a), (b), and (c) as defined in the paper. It's crucial to note that $\boldsymbol{\tilde{\pi}}^i$, the anticipated joint policy, is composed of both the agent's own policy $\pi^{ii}$ and its anticipations of other agents' policies $\pi^{ij}$ (j $\neq$ i). The objective function in Eq. 12 is a function of this entire anticipated joint policy $\boldsymbol{\tilde{\pi}}^i$, not just the agent's own policy. When an agent optimizes this objective, it is simultaneously updating:
> >
> > (1) Its own policy $\pi^{ii}$
> >
> > (2) Its anticipations of other agents' policies $\pi^{ij}$ (j $\neq$ i)
> >
> > The anticipations $\pi^{ij}$ are optimized to maximize the agent's individual advantage function $A^{\pi_{\text{old}}}_i(s,a)$. This means that the anticipated policies reflect what the agent believes other agents should do to maximize its own advantage.
> > The constraint (b) in Eq. 12 ensure that these anticipations don't deviate too far from the actual policies of other agents. This balance between optimizing for individual advantage and maintaining realistic anticipations is key to our method.
> >
> > In the practical implementation (Eq. 16), we use an importance sampling term $\xi_i \cdot \xi_{N_i}$ where $\xi_{N_i}$ is defined in Eq. 13. This term involves the anticipated policies $\pi^{ij}$, further influencing their optimization.
> > Through this process, each agent's anticipations of others' policies come to reflect that agent's preferences, as shaped by its individual reward function, while still remaining grounded in the actual behavior of other agents.
> >
> > We acknowledge that our initial explanation of this process was not sufficiently clear. In our revised paper, we provided a more detailed explanation of how the optimization of anticipated policies occurs and how this relates to individual agent preferences after Eq. 16.

---

> > > ### Author Response · Authors · 2024-08-25
> > > **Author Response to Reviewer ngjK**
> > >
> > > **2. The proposed algorithm of AS is motivated by the setting in which all agents want to optimise a single objective of total agent returns (Eq. 1) but have only access to their own individual rewards since sharing a common reward would not preserve privacy about agents' goals. However, it is unclear to me whether agents sharing anticipated policies, as proposed in AS, with other agents itself preserves privacy...**
> > >
> > > We sincerely appreciate the reviewer's insightful comments regarding privacy preservation in our Anticipation Sharing (AS) method. The reviewer raises a crucial point about potential information leakage through shared anticipated policies, which warrants careful consideration.
> > >
> > > Our work was indeed motivated by scenarios where reward privacy is important. However, we acknowledge that our current approach focuses on avoiding explicit exposure of rewards rather than providing formal privacy guarantees. The reviewer is correct in pointing out that the shared anticipated policies may implicitly reveal information about an agent's rewards or preferences. To clarify our current stance and future directions:
> > >
> > > (1) *Current Approach:* AS qualitatively reduces information sharing compared to methods that directly share rewards or full policies. However, we agree that this protocol does not provide provable privacy guarantees.
> > >
> > > (2) *Trade-off:* In cooperative settings, some level of information sharing about preferences is often necessary for effective coordination. AS attempts to balance this need with privacy concerns, but we acknowledge that this balance requires further investigation.
> > >
> > > (3) *Limitations:* We recognize that our work does not provide quantitative privacy analysis. This is a limitation of our current study and an important area for future research.
> > >
> > > (4) *Future Work:* The reviewer's suggestion for an experiment to quantify potential information leakage is excellent. In future work, we plan to:
> > >
> > > a. Conduct experiments to measure the amount of information inadvertently leaked by sharing anticipations about actual reward functions.
> > >
> > > b. Explore existing techniques (such as differential privacy) and novel approaches to attaining provable and stronger privacy guarantees.
> > >
> > > c. Investigate the trade-off between privacy preservation and cooperative performance.
> > >
> > >
> > > In our revised paper, we stated the limitations of our current approach regarding privacy preservation, including the potential for information leakage, and outlined future research directions for quantitative privacy analysis in this setting in Section 6.

---

> > > > ### Author Response · Authors · 2024-08-25
> > > > **Author Response to Reviewer ngjK**
> > > >
> > > > **Experiments**
> > > >
> > > > **1. Baselines**
> > > >
> > > > We appreciate the reviewer's thoughtful questions regarding our experimental setup and baselines. To address these concerns comprehensively, we'd like to clarify several aspects of our methodology.
> > > > Firstly, we want to emphasize that all baseline algorithms and our AS method are implemented on the foundation of the same PPO-based MARL algorithm. This ensures that any performance differences stem from the information sharing mechanisms rather than underlying algorithm variations. We acknowledge that this crucial detail was not made explicit in our original submission and will add this clarification in our revision.
> > > >
> > > > Regarding the optimization objective, both our AS algorithm and all baseline algorithms are indeed designed to optimize the total return of all agents under individual rewards, as mentioned in Section 5.2. This consistency in objective allows for a fair comparison across methods.
> > > >
> > > > We thank the reviewer for suggesting two additional baselines. We have implemented these algorithms and included the results in Figure 6 in our revised paper. Specifically, we added Independent PPO with individual rewards and Centralized PPO with shared rewards.  For the Independent PPO with individual rewards, each agent learns its own policy using single-agent PPO. For the centralized PPO, we used a single policy network to output probabilities of the joint action distribution and a critic network to evaluate the total return using the total reward. Due to the exponential growth of the joint action space, we limited this experiment to N=2 agents. The results show that Independent PPO achieves the lowest total return, while Centralized PPO performs the best. Our AS algorithm's performance is closest to that of Centralized PPO, demonstrating its effectiveness in balancing individual privacy and collective performance.
> > > >
> > > > Concerning hyperparameter selection, we chose values based on common practices in the field. For instance, we set the discount factor to 0.99 and used the same clipping threshold as in the original PPO paper. Network sizes were determined based on the state and action dimensions of each environment. We selected baseline algorithms that were originally designed to optimize the total return of all agents under individual rewards, ensuring a fair comparison with our method.
> > > >
> > > > We recognize that our initial presentation of these experimental details was not as thorough as it should have been. In our revision, we provided a more comprehensive description of our experimental setup, including these additional baselines and a more detailed explanation of our hyperparameter choices. We believe these additions will offer a clearer and more complete picture of our method's performance in context.
> > > >
> > > >
> > > >
> > > > **2. Training curves. In the Exchange task, the total returns of PS appears to degrade in later stages of training. Why is that?**
> > > >
> > > > Thanks for pointing it out. We are not sure about the reason of the degradation of PS in the Exchange task. We think it might have not converged as we mentioned in the Section of Discussion and conclusions, "PS exhibits slow convergence on some tasks, possibly due to the overhead of sharing entire policy parameters, which may introduce redundant information not essential for effective coordination". Because the training of PS takes a long time, we did not run more episode.
> > > >
> > > > **3. Training metrics**
> > > >
> > > > Thanks for the suggestions. We have conducted experiments that can show the anticipated actions and the discrepancy between the actions chosen by an agent and the anticipated actions given by another agent. We added these experiments in Section 5.3 in our revised paper. Specifically, for ease of understanding, we use the task of Cooperative Predation with two agents. For this task, the action set includes two actions: "moving towards the target" and "moving away from the target". The optimal policy that can maximize the collective total returns is both agent moving towards the target.
> > > >
> > > > In order to know the anticipated actions learned by each agent, we calculate the proportion of the anticipated actions being "moving towards the target". The results are shown in the top row of Figure 7 in our revised paper. The results indicate that both agents anticipate that the other agent can move towards the target rather than move away from the target with a proportion approaching 1.
> > > > The mean square error between the probability of the action chosen by an agent and the anticipated action given by the other agent is shown in the second row of Figure 7 in our revised paper. As the training proceeds, the MSE becomes smaller.

---

> > > > > ### Author Response · Authors · 2024-08-25
> > > > > **Author Response to Reviewer ngjK**
> > > > >
> > > > > **4. Ablation**
> > > > >
> > > > > Thanks for the suggestions. We conducted the ablation study using the cooperative predation and cleanup tasks. The experimental results are shown in Figure 5 in our revised paper. The results demonstrate that when the constraints are removed, the algorithm performance drops or the agents cannot even learn anything.
> > > > >
> > > > >
> > > > > **5. Scalability study**
> > > > >
> > > > > Thank you for your helpful suggestion. In our algorithm, each agent learns its own policy and anticipated policies about other agents. Currently, AS algorithm trains $N^2$ policy networks. For the baseline algorithms, VS and VPS train N policies. For PS where each agent needs to learn a joint policy, we also train $N^2$ policies (N policies each agent) in our experiment for a fair performance comparison.
> > > > >
> > > > > We believe there are potential solutions to reduce the complexity of our AS algorithm. For example, we can use a more computationally efficient network structure, such as using a multi-head policy network that has $N$ output units to output the agent's own policy and $N-1$ anticipated policies. Then, each agent only needs to train one network, and the computational cost will be decreased. Due to the scope of this work, we leave the further study regarding scalability to our future work.  In our revised paper, we explained the computational cost and the potential solution to reduce the cost in Section 6.
> > > > >
> > > > >
> > > > > **6. Sensitivity analysis**
> > > > >
> > > > > Thanks for the comments. The range of the changing weight is related to the value of the weight. As the experimental results in Figure 4 of our paper indicates, the weight is not overly sensitive to precise tuning. But a rough tuning is still necessary.
> > > > > As for the appropriate value of the constraint weight, it might be related to the specific action space, such as the dimension of the action space or the meaning of different action dimensions. We conducted the experiments in Cooperative predation with $\rho=1000$ and Cleanup with $\rho=0.1$. The results show that when the value of $\rho$ is changed too much, the performance of the algorithm will be affected massively. Therefore, a rough tuning of $\rho$ is necessary. In our revised paper, we clarified this point.
> > > > >
> > > > >
> > > > > **Related work**
> > > > >
> > > > > **1. Due to the similarity of the proposed constrained optimization problem and trust region optimization in the multi-agent context, it appears that [6] is significantly related. This work studies trust region bounds in MARL under decentralized PPO. Any discussion on the similarity of the proposed approach and the trust region formulation in [6] would be helpful.**
> > > > >
> > > > > We have cited this reference in the Introduction and Related work. This work focuses on optimizing multi-agent policies under the assumption of a shared team reward, whereas our work aims at solving cooperative MARL under individual rewards.
> > > > >
> > > > > **2. By motivation, the proposed setting appears similar to no-conflict games in which agents have individual rewards but agree on the optimal joint policy. There have been approaches proposed for this setting that leverage this property by optimising agents that assume that other agents will aim to maximise their returns (e.g. [7, 8]). Further discussion of similarity of these settings and potential experiments of these algorithms in your evaluation would be appreciated.**
> > > > >
> > > > > Thanks for the references. References [7] and [8] focus on no-conflict games where  "the agents have the same set of joint policies that maximise their expected returns" [7]. They aim at solving the Pareto-optimal equilibrium in no-conflict games, i.e., a joint policy that can maximise the returns of all agents. However, our work does not focus on the no-conflict games, but the setting where agents have different joint policies that maximise their own returns - agents' rewards conflict with each other. An example is Prisoner's Dilemma. One agent will get the maximal reward with the joint policy (Cooperate, Defect), whereas the other agent will get the maximal reward with the joint policy (Defect, Cooperate). Thus, the joint policies that maximise the two agents' own returns are different. Therefore, it is not a no-conflict game.
> > > > >
> > > > > **Requested Changes**
> > > > >
> > > > > We have addressed all the requested changed in our revised paper. Please find the details in our revised paper or the responses above.

---

> ### Comment · Reviewer_ngjK · 2024-09-03
> **Response to Authors**
>
> I thank the authors for their detailed revisions and responses. Please find follow-up questions below:
>
> # Problem Setting and Examples
>
> The clarifications regarding the setting, its relationship to existing formalisms and literature, and the discussion of conflicts that may arise during training are helpful additions. That being said, I still find the setting somewhat contrived and the newly listed motivational examples odd to formalise in this way.
>
> > (1) Supply Chain Optimization: Multiple companies in a supply chain need to coordinate their production and logistics to maximize overall efficiency. Each company has its own profit function (individual reward) that it wants to keep private due to competitive concerns. However, the overall supply chain performance (collective return) depends on their coordinated actions. Our method allows these companies to cooperate effectively without revealing their individual profit functions.
>
> While each company in this example might have an interest in a working supply chain, fundamentally each company only has an interest in maximising its own profits and controlling the supply chain towards that. Therefore, I am not convinced that this is a reasonable example for your setting but instead each agent (company) should maximise only its own cumulative rewards (profits).
>
> > (2) Distributed Energy Systems: In a smart grid, various energy producers and consumers need to coordinate to balance supply and demand. Each entity has its own cost/utility function (individual reward) that it may not want to disclose. Yet, the stability and efficiency of the entire grid (collective return) depend on their coordinated actions. Our approach enables this coordination while preserving the privacy of individual cost/utility functions.
>
> Similar to the previous example, individual agents (consumers/ producers) within the grid might have an interest in its stability but arguably only to the extend of satisfying their own usage to keep their cost small/ utility large. This would, again, most naturally be reflected by each agent maximising its own cumulative rewards that would also capture each agents' interest in maintaining a stable grid to the extent of the grid satisfying their needs.
>
> > (3) Federated Learning in Healthcare: Multiple hospitals aim to collaboratively train a medical diagnosis model without sharing patient data. Each hospital has its own performance metric (individual reward) based on its specific patient population and priorities. The goal is to create a model that performs well across all hospitals (collective return) without compromising individual hospital data or metrics.
>
> From the perspective of each agent (hospital), they are arguably only interested in obtaining a model that works best on their patient population and not on the population of other agents. That being said, this might be the only slightly compelling example in my opinion since a model that generalises across a larger population might also be of benefit to all agents / hospitals.
>
> On a related notion, when mentioning related literature on no-conflict games, I was primarily wondering whether there might be overlap between no-conflict games and tasks in which your setting might be reasonably applicable. To further illustrate what I mean, your defined setting is arguably not useful in typical competitive zero-sum games in which the rewards of all agents are directly conflicting. For example in chess or Go, where one might define individual rewards of agents as +1 for winning, -1 for losing, and 0 for draw, your formalism would aim for all agents to optimise the sum of all agents cumulative rewards but this sum is always zero. Therefore, your algorithm would always strive for agents to draw which might not be particularly useful
>
> This naturally raises the following question: **In what games is the proposed setting useful/ sensible?** I was wondering whether the answer to this question might be no-conflict games in which, by definition, all agents agree on the optimal joint policy and, thus, would have an incentivise in optimising towards the optimisation objective defined in your work in Eq. 1. Your example of prisoners dilemma suggests that there are games that might benefit from your setting and are not no-conflict games but that still does not represent a clear answer to the posed question which I believe is important to answer. By providing a clear answer to this question, you might get closer to clarity on the proposed setting and its potential applications.
>
> # Theory Contributions
>
> Re Eq. 23, I have two follow up comments/ questions:
> 1. The expectation in the 2nd last line should be written more explicitly as $s_0 \sim d(s_0)$.
> 2. How do you get from line 3 to line 4? Why is $\sum_{i=0}^N \sum_{t=0}^\infty \gamma^{t+1} V_i^{\pi_{old}}(s_{t+1}) = \sum_{i=0}^N \sum_{t=0}^\infty \gamma^t V_i^{\pi_{old}}(s_t)$?
>
> The remaining revisions look good to me and significantly improve the clarity of this work.

---

> > ### Author Response · Authors · 2024-09-04
> > **Author Response to Reviewer ngjK**
> >
> > Thank you for your insightful comments. Below please find our point by point replies.
> >
> > **Problem Setting and Examples**
> >
> > *Reviewer: "I still find the setting somewhat contrived and the newly listed motivational examples odd to formalise in this way."*
> >
> > Response: We appreciate your concern about the examples. Our work focuses on a subset of cooperative multi-agent scenarios where agents need to cooperate to maximise the collective return while having conflicting and private individual rewards. Specifically, we focus on social dilemma problems, as presented in our Introduction, Related Work, and Results sections.
> >
> > *Reviewer: "While each company in this example might have an interest in a working supply chain, fundamentally each company only has an interest in maximising its own profits and controlling the supply chain towards that."*
> >
> > Response: We fully acknowledge that companies primarily aim to maximise their own profits, as you rightly point out. However, in supply chains, this individual profit maximisation can lead to suboptimal outcomes for all participants due to phenomena like the "bullwhip effect". This represents a classic social dilemma where short-term individual optimisation leads to long-term collective (and individual) losses. Our method doesn't require companies to compromise individual profits. Instead, it allows them to discover strategies that maximise their long-term individual profits by considering the broader supply chain dynamics.
> >
> > *Reviewer: "Similar to the previous example, individual agents (consumers/ producers) within the grid might have an interest in its stability but arguably only to the extent of satisfying their own usage to keep their cost small/ utility large."*
> >
> > Response: We agree that agents primarily focus on their own costs/utilities. However, in energy systems, individual optimisation without coordination can lead to grid instability, which ultimately affects all participants negatively. This scenario represents a social dilemma where purely self-interested behaviour may lead to the worst outcomes for all agents in the long run. Our approach enables agents to cooperate towards grid stability while maintaining privacy, serving their own interests through a more reliable and potentially cost-effective system.
> >
> > *Reviewer: "From the perspective of each agent (hospital), they are arguably only interested in obtaining a model that works best on their patient population and not on the population of other agents."*
> >
> > Response: You're right that hospitals prioritise their own patient populations. However, certain medical challenges benefit significantly from larger, diverse datasets. Individual optimisation (focusing only on one's own data) leads to suboptimal outcomes. Our method facilitates collaborative improvement of models while protecting each hospital's sensitive data and competitive edge, potentially enhancing their service quality for complex cases.
> >
> > *Reviewer: "On a related notion, when mentioning related literature on no-conflict games, I was primarily wondering whether there might be overlap between no-conflict games and tasks in which your setting might be reasonably applicable."*
> >
> > Response: Our setting is distinct from no-conflict games. Unlike no-conflict games, our setting involves scenarios where agents' immediate interests may conflict. The challenge is to discover cooperative strategies that yield better long-term outcomes for all, despite these initial conflicts.
> >
> > *Reviewer: "Your defined setting is arguably not useful in typical competitive zero-sum games in which the rewards of all agents are directly conflicting."*
> >
> > Response: You are correct. Our approach is indeed not designed for zero-sum games such as chess or Go. Our AS algorithm is tailored for scenarios where cooperation can enhance outcomes for all participants, which is inherently not possible in strictly competitive zero-sum settings.

---

> ### Author Response · Authors · 2024-09-04
> **Author Response to Reviewer ngjK**
>
> *Reviewer: "In what games is the proposed setting useful/ sensible?"*
>
> Response: Our setting is particularly useful for a subset of general-sum games that exhibit characteristics of social dilemmas. Specifically, it applies to scenarios where: a) Short-term individual optimisation leads to suboptimal long-term outcomes for all agents (similar to the Prisoner's Dilemma). b) There exists a cooperative solution that yields better long-term outcomes for all agents, but this solution is not immediately apparent or achievable through individual optimisation. c) Agents are unwilling or unable to share their reward functions due to privacy concerns or competitive reasons.
>
> As you noted, the Prisoner's Dilemma is a classic example, but our work extends to more complex, real-world scenarios. For reference, Table 2 in Christianos et al. (2023) (reference [8] in your previous comments) provides a useful categorisation of MARL environments, including no-conflict, zero-sum, and social dilemma settings.
>
> We acknowledge that applying theoretical models to real-world scenarios presents significant challenges, particularly in complex multi-agent systems where individual incentives may not always align with collective benefits. Our current work provides a foundation for addressing these challenges, but we recognise that further research is needed to bridge the gap between theory and practice. Future work could explore the integration of traditional incentive structures with our AS framework to further encourage cooperation in settings where personal profit is a primary driver. Additionally, we aim to conduct more extensive empirical studies in diverse real-world environments to refine and validate our approach. These potential enhancements could make our model more flexible and applicable across a range of real scenarios, addressing some of the practical challenges inherent in multi-agent cooperation.
>
> We hope these clarifications better illustrate the unique aspects of our problem setting and its relevance to scenarios where balancing individual interests, collective benefits, and privacy is crucial. We have also updated a newly revised paper where we further clarified the motivational examples in the Introduction.
>
>
> **Theory Contributions**
>
> *Reviewer: Re Eq. 23, I have two follow up comments/ questions:*
>
> *1. The expectation in the 2nd last line should be written more explicitly as $s_0 \sim d(s_0)$.*
>
> *2. How do you get from line 3 to line 4? Why is $\sum_{i=0}^N \sum_{t=0}^\infty \gamma^{t+1} V_i^{\pi_{old}}(s_{t+1}) = \sum_{i=0}^N \sum_{t=0}^\infty \gamma^t V_i^{\pi_{old}}(s_{t})$.*
>
> Response:
>
> 1. Thank you for pointing that out. We have added $\sim d(s_0)$ in the 2nd last line of Eq. 23 in our updated revised paper.
>
> 2. Thank you for your follow-up question. Please note that in line 4, it is $\sum_{i=0}^N \sum_{t=1}^\infty \gamma^t V_i^{\pi_{old}}(s_{t})$, i.e., the time variable starts from 1 rather than 0. Therefore, from line 3 to line 4, we get $\sum_{i=0}^N \sum_{\bf{t=0}}^\infty \gamma^{t+1} V_i^{\pi_{old}}(s_{t+1}) = \sum_{i=0}^N \sum_{\bf{t=1}}^\infty \gamma^t V_i^{\pi_{old}}(s_{t})$.

---

> > ### Comment · Reviewer_ngjK · 2024-09-06
> > **Response to Authors**
> >
> > I thank the authors for their clarifications.
> >
> > > Specifically, we focus on social dilemma problems, as presented in our Introduction, Related Work, and Results sections.
> >
> > Could the authors point to a particular definition of social dilemma problems. What *exactly* constitutes a social dilemma? In your response, you write that within the games your work focuses on, the following holds:
> >
> > > There exists a cooperative solution that yields better long-term outcomes for all agents, but this solution is not immediately apparent or achievable through individual optimisation.
> >
> > However, if there is a cooperation solution = joint policy that achieves optimal returns for all agents (according to their individual reward functions), then these games are by definition no-conflict games since no-conflict games are merely defined by the fact that all agents (as per their individual reward functions) prefer the same joint policy across all agents. Furthermore, this seems to contradict the statement below made in your response:
> >
> > > Our setting is distinct from no-conflict games. Unlike no-conflict games, our setting involves scenarios where agents' immediate interests may conflict. The challenge is to discover cooperative strategies that yield better long-term outcomes for all, despite these initial conflicts.
> >
> > There might be a misunderstanding about the definition of these settings and I'd appreciate a precise response since I believe it is of great importance to understand in what problems your work might be applicable.
> >
> > I thank the authors for their clarifications regarding the theory contributions, these look good to me.

---

> > > ### Author Response · Authors · 2024-09-06
> > > **Author Response to Reviewer ngjK**
> > >
> > > Thank you for your comments.
> > >
> > > **Reviewer: Could the authors point to a particular definition of social dilemma problems. What exactly constitutes a social dilemma?**
> > >
> > > Response: Social dilemma is a common problem in many areas including social science, behavior science, economics, biology, etc. The definition of social dilemma can be found in Shen et al. (2023) as "Social dilemma games model scenarios where individual benefits may conflict with collective benefits, and cooperation is necessary for collective well-being", and in Kollock et al. (1998) as "Social dilemmas are situations in which individual rationality leads to collective irrationality. That is, individually reasonable behavior leads to a situation in which everyone is worse off than they might have been otherwise", and in Dawes et al. (1980) as "Social dilemmas are situations in which each member of a group has a clear and unambiguous incentive to make a choice that when made by all members provides poorer outcomes for all than they would have received if none had made the choice."
> > >
> > > Social dilemma is a problem caused by the conflicting individual rewards of agents, where "optimizing rewards based solely on individual interests often leads to suboptimal collective outcomes and, consequently, suboptimal individual outcomes" as we introduced in our paper.
> > >
> > > **Reviewer: However, if there is a cooperation solution = joint policy that achieves optimal returns for all agents (according to their individual reward functions), then these games are by definition no-conflict games since no-conflict games are merely defined by the fact that all agents (as per their individual reward functions) prefer the same joint policy across all agents. ... There might be a misunderstanding about the definition of these settings.**
> > >
> > > Response:
> > > Social dilemmas are different from no-conflict games in reward structure. For simplicity, we can take Prisoner's dilemma for example. The table below shows an example of a reward setup, where "C" represents "Cooperate" and "D" represents "Defect".
> > >
> > > |   | C     | D     |
> > > |---|-------|-------|
> > > | C | -1,-1 | -4,0  |
> > > | D | 0,-4  | -3,-3 |
> > >
> > > This is a typical social dilemma problem, as when each agent optimises its own reward, they will get the Nash equilibrium, (D, D). However, there exists a cooperative solution (C, C) that yields better outcomes for all agents compared with (D, D). This is a detailed example of our description "There exists a cooperative solution that yields better long-term outcomes for all agents, but this solution is not immediately apparent or achievable through individual optimisation". Meanwhile, this example also demonstrates that social dilemma is distinct from no-conflict games. As can be seen in this example, the optimal joint policy for the row agent is (D, C), whereas the optimal joint policy for the column agent is (C, D). These two joint policies are different, which indicates the conflict between agents' individual interests. We believe that our explanation is aligned with the definitions of social dilemma and no-conflict game.
> > >
> > >
> > > **References**
> > >
> > > Shen, Chen, et al. "How committed individuals shape social dynamics: A survey on coordination games and social dilemma games." Europhysics Letters 144.1 (2023): 11002.
> > >
> > > Kollock, Peter. "Social dilemmas: The anatomy of cooperation." Annual review of sociology 24.1 (1998): 183-214.
> > >
> > > Dawes, Robyn M. "Social dilemmas." Annual review of psychology (1980).

---

### Review · Reviewer_sWkt · 2024-08-09

**Summary Of Contributions:**

In this work, the authors introduce Anticipation Sharing, a de-centralized multi-agent RL algorithm for finding joint policies with high total welfare. Communication is limited to agent’s anticipations about other agents’ action distribution at visited states, rather than complete policies or the agent’s private reward function. The authors provide results on a few prior domain that require agent cooperation, as well as a couple of new domains.

**Audience:**

Yes

**Claims And Evidence:**

No

**Requested Changes:**

*Larger issues*

Who would actually be running this, a centralized controller, or individual agents? There seems to be some internal conflict within the motivation and setup.
A centralized controller might want to find a policy that optimizes some measure of common welfare, but an individual agent can have quite a strong incentive to have a different policy. This is especially true for total welfare, which might come from sacrificing a few agents who might reasonably object. If this is actually run by competitive agents in a distributed manner, I think agents have an incentive to lie in their communication.
An individual might be concerned about keeping their rewards private, but a centralized controller finding a joint policy would prefer more information.
Or re-stated a different way, section 4 states “In cooperative MARL settings with individual rewards, agents must balance personal objectives with collective goals”. Why must an individual agent balance things? Why would an agent participate in a process that reduces their utility?
The paper needs to make the answer to these questions clear.

Thm 2:  sum_s,a(~pi(a|s) - pi(a|s))^2
Is this really a sum over all possible state action pairs? This is large even in finite state spaces, and unbounded (undefined?) for continuous state spaces. Given it’s tested on a continuous domain, this is a bit of a problem.

“By maximizing this lower bound…”
This argument is technically correct but it does not follow that it produces a useful outcome. Consider the trivial bound
n(new) >= n(old) - HUGE + z(s,a)
It is not practical to say we can ignore n(old) and HUGE which don’t depend on new, and just optimize n(new) by maximizing z(s,a). For sufficiently large values of HUGE, I can use any z(s,a).

Suggest the authors avoid directly saying the method is “privacy preserving” or that is “preserving privacy”. That has been a formal area of study for long enough that those phrases are implying something more like “outside agents can’t recover information X” rather than just meaning “an agent does not have to directly reveal some information X”.

Figure 2
Taking a brief look at the Zhang et al. 2018b, the VPS results for C. Navigation results don’t seem to match Figure 2. Assuming it’s still 20 agents in this paper, there’s a factor of 20 for total (this paper) vs average (VPS paper) but that’s still not enough to match. This was the first and easiest to check – can the authors explain the discrepancy? What am I missing?
Similar questions about cleanup and harvest. What are the differences from Figure 8 of Chistoffersen et al. and what’s being reported in this work? Different scales or values being reported? Environment setup?  Just different (much less effective?) baseline algorithm choices? Figure 8 reports welfares around 400-650 in Harvest and 200-600 in Cleanup, compared to total return of 15-25 for Harvest and 1-2 for Cleanup in the Figure 2 of this paper.


*Smaller changes*

r^i_t = R^i(s,a) – are deterministic rewards needed, or could this immediately be extended to sampled values?

“collective cumulative return”  -> “collective cumulative average return”?
Given s0 ~ d^pi(s), the paper is using the average over visited states, this might be a more appropriate description.

d^pi(s) should be defined

Thm 1 and Thm 2 should probably be moved to the appendix, as lemmas. They don’t seem to be adding anything to the rest of the paper other than their combination directly leads to Thm 3.

Algorithm 1  “for i=1 to N do”
Is this for loop intended to be parallelization across agents? if so, suggest using different terminology, like  “for each agent i do” or something like that, to note that this is not one agent iterating over a variable i and doing something.

Step 2 “Simply optimizing the advantage function may not sufficiently increase these discrepancies”.
Why not?  Text should explain.

**Strengths And Weaknesses:**

The proposed AS algorithm does not require sharing complete policies, or private rewards. Informally, the motivation of agents wanting to keep rewards private is satisfied, to the extent that they are not required to directly broadcast their rewards.

There are however a few substantial concerns noted below, include the motivation, the theoretical argument, and the experimental results.

---

> ### Author Response · Authors · 2024-08-25
> **Author Response to Reviewer sWkt**
>
> **Larger issues**
>
> **1. Who would actually be running this, a centralized controller, or individual agents? There seems to be some internal conflict within the motivation and setup. A centralized controller might want to find a policy that optimizes some measure of common welfare, but an individual agent can have quite a strong incentive to have a different policy. This is especially true for total welfare, which might come from sacrificing a few agents who might reasonably object. If this is actually run by competitive agents in a distributed manner, I think agents have an incentive to lie in their communication. An individual might be concerned about keeping their rewards private, but a centralized controller finding a joint policy would prefer more information. Or re-stated a different way, section 4 states “In cooperative MARL settings with individual rewards, agents must balance personal objectives with collective goals”. Why must an individual agent balance things? Why would an agent participate in a process that reduces their utility? The paper needs to make the answer to these questions clear.**
>
> Thank you for these important questions about our system. We appreciate the opportunity to clarify:
>
> (1) *System operation:* Our system is designed for individual agents, not a centralized controller. Each agent operates independently, making decisions based on its own information and anticipations about others. Such decentralised setting is common and necessary in scenarios where agents are distributed and cannot learn and be controlled with a central unit.
>
> (2) *Cooperation under individual rewards:* Our work focuses on scenarios where agents need to cooperate to maximize their collective performance, even when they only have access to their individual rewards. This is common in team-based tasks where the overall outcome depends on collaborative effort.
>
> (3) *Balancing goals:* Agents need to consider collective goals rather than only maximizing their own rewards even though they only know their own individual rewards and learn policies in a decentralised way. As illustrated in the Prisoner's Dilemma, when agents act purely out of self-interest, it can lead to lower total returns compared to cooperating for the common good.
>
> (4) *Privacy considerations:* In our setup, agents prioritize the confidentiality of their policies, rewards, and values. This is to mitigate risks such as malicious attacks, unwanted disclosure of strategic interests, or potential loss of intellectual property.
>
> (5) *Anticipation Sharing method:* Our approach allows agents to exchange anticipated action distributions instead of directly sharing sensitive information like rewards or full policies. This method aims to enable cooperation while maintaining a degree of privacy.
>
>
> **2. Thm 2: $\sum_{s,a}(\tilde{\pi}(a|s) - \pi(a|s))^2$. Is this really a sum over all possible state action pairs? This is large even in finite state spaces, and unbounded (undefined?) for continuous state spaces. Given it’s tested on a continuous domain, this is a bit of a problem.**
>
> Yes, it is a sum over state and action pairs. For continuous state spaces, it will be an integral. In the algorithm we proposed, the objective of policy learning, Eq. 16, involves minimizing this term. In practice, the algorithm minimizes the discrepancy between $\tilde{\pi}(a|s)$ and  $\pi(a|s)$ based on the visited states.
>
> Thank you for this crucial observation regarding the summation over state-action pairs in Theorem 2. For discrete state and action spaces, the summation $\sum_{s,a} (\boldsymbol{\tilde{\pi}}^i(a|s) - \boldsymbol{\pi}(a|s))^2$ is well-defined. However, for continuous spaces, this should be reformulated as an integral. For the integral to be meaningful, we would need to establish conditions on $\boldsymbol{\tilde{\pi}}^i$ and $\boldsymbol{\pi}$ to ensure the integrand is integrable over $\mathcal{S} \times \mathcal{A}$. In our practical implementation (Eq. 16), we approximate this using the expectation over states: $E_{s} \sum_{a \in \mathcal{A}} (\boldsymbol{\tilde{\pi}}^i(a|s) - \boldsymbol{\pi}(a|s))^2$.
>
> It should be noted that in practice, our algorithm minimizes the discrepancy between $\boldsymbol{\tilde{\pi}}^i(a|s)$ and $\boldsymbol{\pi}(a|s)$ based only on the states visited during training. This approach is common in reinforcement learning to make the computation tractable (Van Hasselt, Hado. "Reinforcement learning in continuous state and action spaces."), but it introduces a gap between our theoretical bound and practical implementation. We leave the analysis regarding the effect of this gap on the performance of our algorithm to our future work.

---

> > ### Author Response · Authors · 2024-08-25
> > **Author Response to Reviewer sWkt**
> >
> > **3. “By maximizing this lower bound…” This argument is technically correct but it does not follow that it produces a useful outcome. Consider the trivial bound n(new) >= n(old) - HUGE + z(s,a) It is not practical to say we can ignore n(old) and HUGE which don’t depend on new, and just optimize n(new) by maximizing z(s,a). For sufficiently large values of HUGE, I can use any z(s,a).**
> >
> > We appreciate your concern about the validity of ignoring certain terms in our optimization process. You are correct that in general, maximizing a lower bound doesn't necessarily lead to optimizing the original objective, especially when large terms are ignored. However, in our case, we derive the surrogate objective in Section 4.2 from Eq. 10 by ignoring the terms $\eta(\pi_\text{old}) - f^{\pi_\text{old}}$, where $f^{\pi}$ is defined as Eq.8 in our paper.
> >
> > We acknowledge that we should have provided a more rigorous justification for this step. Our rationale is as follows:
> >
> > (1) We focus on scenarios with finite and relatively small action spaces, which are common in many real-world applications, so $|\mathcal{A}|$ (the size of the action space) is not excessively large.
> >
> > (2) The term $\|d^{\pi_\text{old}}\|^2_2$ is the squared L2-norm of the state visitation distribution, which is bounded.
> >
> > (3) The advantage function $A^{\pi_\text{old}}_i(s,a)$ is also bounded as the reward value is bounded.
> >
> > Given these considerations, we expect $\eta(\pi_\text{old}) - f^{\pi_\text{old}}$ to be of reasonable magnitude in our target scenarios.
> > In our revised paper, we provided a more detailed analysis of the magnitude of the ignored terms in Section 4.2.
> >
> >
> > **4. Suggest the authors avoid directly saying the method is “privacy preserving” or that is “preserving privacy”. That has been a formal area of study for long enough that those phrases are implying something more like “outside agents can’t recover information X” rather than just meaning “an agent does not have to directly reveal some information X”.**
> >
> > We appreciate the reviewer's insightful comment regarding using the terms *privacy preserving* and *preserving privacy*. We acknowledge that these terms have specific connotations in formal privacy research that our work does not fully address.
> >
> > Our work was motivated by scenarios where agents wish to avoid explicitly sharing their reward functions. Our method does reduce the amount of information shared between agents compared to approaches that directly share rewards or full policies. However, we do not provide formal privacy guarantees in the sense of differential privacy or other quantitative privacy measures.
> >
> > In light of your comment, we agree that our privacy-related terminology should be more precise. In our revision, we clarified our method with more accurate descriptions such as *reward-private* and *designed primarily for cooperation without explicitly sharing rewards, values or policies*. We also added a discussion about privacy preserving issue in the last section of our revised paper.
> >
> > It is thus noted that the concern of keeping rewards private has motivated our work, but we recognize that this paper does not tackle the problem of developing MARL protocols and algorithms to meet the requirement of preserving reward privacy under explicit and quantitative constraints. The problem is out of the scope of this paper. To the best of our knowledge, there has not been related work involving multi-agent cooperation under private rewards, and our work represents a first step in this direction of MARL. We will leave the quantitative study on privacy preservation to future work.

---

> > > ### Author Response · Authors · 2024-08-25
> > > **Author Response to Reviewer sWkt**
> > >
> > > **5. Figure 2 Taking a brief look at the Zhang et al. 2018b, the VPS results for C. Navigation results don’t seem to match Figure 2. Assuming it’s still 20 agents in this paper, there’s a factor of 20 for total (this paper) vs average (VPS paper) but that’s still not enough to match. This was the first and easiest to check – can the authors explain the discrepancy? What am I missing? Similar questions about cleanup and harvest. What are the differences from Figure 8 of Chistoffersen et al. and what’s being reported in this work? Different scales or values being reported? Environment setup? Just different (much less effective?) baseline algorithm choices? Figure 8 reports welfares around 400-650 in Harvest and 200-600 in Cleanup, compared to total return of 15-25 for Harvest and 1-2 for Cleanup in the Figure 2 of this paper.**
> > >
> > > Thank you for your valuable comments. The discrepancies you've pointed out for Cooperative Navigation, Cleanup, and Harvest are indeed significant and warrant a thorough explanation. Several factors contribute to these differences, including variations in environment parameters, implementation details of baseline algorithms, or evaluation metrics.
> > >
> > > For Cooperative Navigation, we set the time horizon of an episode as 100 time steps. The environment size is $5 \times 5$. We use three agents. Figure~5(a) in the Appendix of our paper illustrated this environment setup.
> > > In Zhang et al. 2018b, we did not find the time horizon. Besides, we implement the baseline algorithms based on PPO algorithm for fair comparison. Zhang et al. 2018b is based on vanilla policy gradient algorithm, which does not use PPO surrogate objective and GAE returns. In addition, Figure 2 in Zhang et al. 2018b shows the average return whereas we use total return.
> > >
> > > For harvest and cleanup, we used less episode time horizon and smaller environment size than those used in the experiments of Chistoffersen et al. https://github.com/Algorithmic-Alignment-Lab/contracts. In Chistoffersen et al., the episode time steps is set as 15000000, and the environment size is $16 \times 38$ in Harvest, and $25 \times 18$ in Cleanup. We set episode time step as 100 and environment size as $7 \times 38$ in Harvest, and $11 \times 18$ in Cleanup. Besides, the algorithm used in Chistoffersen et al. is not based on PPO.
> > >
> > > In our revision, we carefully reviewed our implementation and experimental setup for each environment. We added the corresponding details in Section 5.1.
> > >
> > >
> > > **Smaller changes**
> > >
> > > **1. $r^i_t = R^i(s,a)$ – are deterministic rewards needed, or could this immediately be extended to sampled values?**
> > >
> > > No, deterministic rewards are not necessary for our method. The proposed approach can be applied to settings with both deterministic and stochastic reward functions, as our theories and algorithm have no restriction on the form of the reward function. We use $r^i_t = R^i(s, a)$ for simplicity of notation, but this can be easily extended to sampled values without affecting the core of our method. In our revised paper, we clarified this point in Section 3.
> > >
> > >
> > > **2. “collective cumulative return” -> “collective cumulative average return”? Given $s_0 \sim d^\pi(s)$, the paper is using the average over visited states, this might be a more appropriate description.**
> > >
> > > Thank you for this suggestion. While collective cumulative average return could be considered, we believe collective cumulative return is still appropriate. Our formulation represents the expectation of the collective cumulative return over the initial state distribution, which is indeed a cumulative return, not an average. In our revised paper, we modified "collective cumulative return" as "the expectation of collective cumulative return".
> > >
> > >
> > > **3. $d^\pi(s)$ should be defined.**
> > >
> > > We apologize for this oversight. You are correct that we should have defined $d^\pi(s)$. In our revised paper, we explicitly defined $d^\pi(s)$ as the state visitation distribution under policy $\pi$. We also corrected the notation for the initial state distribution to $s_0 \sim d(s_0)$ on Page 4 and Page 16, and revise Eq. 22 as $\eta(\boldsymbol{\pi})= \sum_{i=1}^N \mathbb{E}_{s_0 \sim d(s_0)} \left[V_i^{\boldsymbol{\pi}} (s_0)\right]$.

---

> > > > ### Author Response · Authors · 2024-08-25
> > > > **Author Response to Reviewer sWkt**
> > > >
> > > > **4. Thm 1 and Thm 2 should probably be moved to the appendix, as lemmas. They don’t seem to be adding anything to the rest of the paper other than their combination directly leads to Thm 3.**
> > > >
> > > > We appreciate this suggestion for improving the paper's structure. After careful consideration, we prefer to keep Theorems 1 and 2 in the main text, but we relabeled them as Lemmas in our revised paper. This decision is based on our belief that maintaining these results in the main body of the paper provides a more complete and coherent explanation of our theoretical framework.
> > > >
> > > >
> > > > **5. Algorithm 1 “for i=1 to N do” Is this for loop intended to be parallelization across agents? if so, suggest using different terminology, like “for each agent i do” or something like that, to note that this is not one agent iterating over a variable i and doing something.**
> > > >
> > > > Thank you for pointing this out. For the first for loop, it can be modified as a parallelization across agents. We have changed this to ``for each agent $i$ do'' in our revised paper.
> > > > For the second for loop, agents update their policies sequentially to avoid the non-stationarity issue. Specifically, after an agent updates policies, it will share the newest distribution of its own action and anticipated actions to its neighbours for them to use to update policies.
> > > >
> > > >
> > > > **6. Step 2 “Simply optimizing the advantage function may not sufficiently increase these discrepancies”. Why not? Text should explain.**
> > > >
> > > > We apologize for not providing a clear explanation for this statement. In our revised paper, we elaborated on why optimizing the advantage function alone may not sufficiently increase these discrepancies. Specifically, if $\hat{A}_i > 0$, according to the main objective function, Eq. 12, the gradient used to update $\pi^{ij}$ will be positive and will lead to the increase of $\pi^{ij}$. If $\frac{\pi^{ij}(a|s, \theta^{ij})}{\pi^{jj}(a|s)} < 1$, i.e. $\pi^{ij}(a|s, \theta^{ij}) < \pi^{jj}(a|s)$, then the gradient caused by the main objective will decrease the discrepancy between $\pi^{ij}$ and $\pi^{jj}$. Therefore, we don't impose the MSE constraint on $\pi^{ij}$. Then, we propose Eq. 14. That is, if $\hat{A}_i > 0$ and $\frac{\pi^{ij}(a|s, \theta^{ij})}{\pi^{jj}(a|s)} < 1$, or $\hat{A}_i < 0$ and $\frac{\pi^{ij}(a|s, \theta^{ij})}{\pi^{jj}(a|s)} > 1$, the (s, a) pair is not included in $X^{ij}$ and thus the constraint on $\pi^{ij}$ is not activated. As we explained in the paper "we define state-action sets $X^{ij}$ to identify where the policy update driven by the advantage exacerbates the discrepancies between the resulting anticipated policies and other agents' current policies, and $X^{ii}$ to identify the discrepancies between the resulting agent's own policy and the ones anticipated from other agents". We have also added more detailed explanations in Section 4.3.

---

### Review · Reviewer_XzaV · 2024-08-11

**Summary Of Contributions:**

This paper studies multi-agent systems in which the agents want to cooperate while preserving their privacy. An approach, called "Anticipation Sharing" (AS), is proposed. The idea is that each agent shares information about their future actions (or action distribution) instead of sharing information about their reward for instance. The contributions are two-fold. On the theoretical side, bounds are proved to connect the improvement in total returns to the advantages (in the sense of RL) of the agents when switching from an old policy to a new policy. In particular, Theorem 3 provides a foundation for local updates based on AS. On the numerical side, a practical algorithm is proposed and tested on several examples of cooperative MARL tasks.

**Audience:**

Yes

**Broader Impact Concerns:**

As pointed out in the paper, the paper is related to privacy-preserving methods, and it could also have an impact on important problems of management of common goods. Such considerations could be discussed in a paragraph on broader impact.

**Claims And Evidence:**

Yes

**Requested Changes:**

It would be interesting to discuss the (theoretical) complexity of the algorithm.

In (11), the first constraint is independent of the object over which we optimize. Should $\pi^{ii}$ be $\tilde\pi^{ii}$ (or even $\tilde\pi^{ii}_{new}$)? Please check the notation for the whole equation (subscript "new" or not, etc.).

Similarly, In (12), the optimization is over $\tilde{\boldsymbol{\pi}}^i$ but it is totally absent from the constraint, which is strange.

**Strengths And Weaknesses:**

**Strengths:** The paper is well written. The intuition behind the algorithm relies on a theoretical foundation. The experiments are done on various examples and the scalability (in terms of number of agents) is also discussed. The performance of the proposed method is compared with several baselines.

**Weaknesses:** There is not much discussions about the complexity of the algorithm.

---

> ### Author Response · Authors · 2024-08-25
> **Author Response to Reviewer XzaV**
>
> **1. It would be interesting to discuss the (theoretical) complexity of the algorithm.**
>
> Thank you for your helpful suggestion.
> In our algorithm, each agent learns its own policy and anticipated policies about other agents. Currently, AS algorithm trains $N^2$ policy networks. The space complexity is $O(N^2)$. However, we believe there are potential solutions to reduce the complexity. For example, we can use a more computationally efficient network structure, such as using a multi-head policy network that has $N$ output units to output the agent's own policy and $N-1$ anticipated policies. Then, the complexity will be $O(N)$. Besides, the time complexity of current AS algorithm is O(N). Due to the scope of this work, we leave the study of reducing algorithm complexity to our future work. We added a discussion regarding this point in the section of Discussion and conclusions in our revised paper.
>
> **2. In (11), the first constraint is independent of the object over which we optimize. Should $\pi^{ii}$ be $\tilde{\pi}^{ii}$ (or even $\tilde{\pi}^{ii}_{new}$)? Please check the notation for the whole equation (subscript "new" or not, etc.).**
>
> Thank you for your careful review and for pointing out this inconsistency. You are correct that there was an error in our notation. We have corrected Equation 11 in our revised paper to address this issue. In the revised equation:
>
> (1) We removed the 'new' subscript from $\boldsymbol{\tilde{\pi}}^i$ in the expectation term of the objective function.
>
> (2) We kept $\pi^{ii}$ in the first constraint, as it refers to the individual policy of agent i, which is part of the anticipated joint policy $\boldsymbol{\tilde{\pi}}^i$.
>
> The first constraint is indeed imposed on $\pi^{ii}$, not $\tilde{\pi}^{ii}$. This is because $\pi^{ii}$ represents the actual policy of agent i, while $\boldsymbol{\tilde{\pi}}^i$ represents the anticipated joint policy, which includes $\pi^{ii}$ and the agent's anticipations of other agents' policies.
> The effect of this constraint on the optimization is to limit how much each agent's individual policy can deviate from its previous policy. This indirectly affects the anticipated joint policy $\boldsymbol{\tilde{\pi}}^i$, as $\boldsymbol{\tilde{\pi}}^i$ is composed of $\pi^{ii}$ and the anticipations of other agents' policies.
>
> We apologize for any confusion this may have caused and thank you for bringing it to our attention. We have revised the notations in our revised paper and also further clarified the the definition of $\boldsymbol{\tilde{\pi}}^i$ and $\pi^{ii}$ after Eq. 12.
>
>
> **3. Similarly, In (12), the optimization is over $\boldsymbol{\tilde{\pi}}^i$ but it is totally absent from the constraint, which is strange.**
>
> Thank you for this insightful observation regarding the notation in Eq 12. Let us clarify. In Eq. 12, while $\boldsymbol{\tilde{\pi}}^i$ doesn't appear explicitly in the constraints, it is implicitly present. As defined in Definition 1, where $\pi^{ii}$ is the agent's own policy and $\pi^{ij}$ (for $j \neq i$) are its anticipations of other agents' policies.
>
> The constraints in Eq. 12 are imposed on $\pi^{ii}$ and $\pi^{ij}$ ($j \neq i$), which together compose $\boldsymbol{\tilde{\pi}}^i$. Therefore, these constraints effectively limit the space of possible $\boldsymbol{\tilde{\pi}}^i$ by constraining its components. Constraint (a) limits how much the agent's own policy can change, while constraints (b) and (c) ensure that the anticipations are close to the actual policies of other agents.
>
> We have revised the paper to clarify this point and ensure that the role of $\boldsymbol{\tilde{\pi}}^i$ in the optimization problem is more clearly explained. Please see the paragraph after Eq. 12.
>
>
> **4. As pointed out in the paper, the paper is related to privacy-preserving methods, and it could also have an impact on important problems of management of common goods. Such considerations could be discussed in a paragraph on broader impact.**
>
> We appreciate the reviewer's suggestion and agree on its importance. In our revised paper, we added a discussion about privacy-preserving issues in the last section. We discussed the limitations of our method and outlined future research directions, including developing more robust privacy guarantees.
> This discussion provides a balanced view of our work's potential societal implications and situate it within the broader landscape of privacy-preserving MARL.
>
> Besides, in the revised Introduction, we introduced some practical scenarios with potential applications in privacy-critical domains like healthcare and industrial collaborations. We also explored the implications for managing common resources such as in smart grid systems, where our approach could facilitate more efficient and fair distribution.

---

### Author Response · Authors · 2024-08-25
**Thank you for your reviews**

The authors sincerely thank all reviewers for their valuable and helpful comments and suggestions. Your efforts have greatly improved the quality of this paper. We have taken your feedback very seriously and have made substantial revisions to address all comments and suggestions. We have provided point-by-point responses to each reviewer's questions and have updated our revised paper, with all changes highlighted in blue for easy reference.

Key improvements include:

* Providing more appropriate and compelling motivating examples that better demonstrate the need for both cooperation and reward privacy (Introduction).

* Clarifying the problem setting and its uniqueness, particularly in relation to existing frameworks like Stochastic Games and Dec-POMDPs (Section 3).

* Improving explanations of the anticipation sharing algorithm (Section 4).

* Correcting and clarifying our theoretical contributions, with expanded proofs and explanations (Section 4 and Appendix).

* Clarifying the implementation details of baseline algorithms and their optimization objectives (Section 5.2).

* Adding new experiments and baselines, including Independent PPO and Centralized PPO, to provide a more comprehensive evaluation of our method (Section 5.3).

* Conducting and presenting results of an ablation study to demonstrate the importance of constraints in our objective function (Section 5.3).

* Providing a more detailed analysis of anticipated actions and policy discrepancies (Section 5.3).

* Addressing concerns about privacy preservation by acknowledging current limitations and outlining future research directions for quantitative privacy analysis (Section 6).

* Discussing the computational costs and potential solutions to reduce complexity in our approach (Section 6).

We believe these revisions have significantly strengthened our paper, addressing the main concerns raised while maintaining and clarifying the novel contributions of our work. We appreciate the opportunity to improve our manuscript and hope that these changes satisfactorily address the reviewers' feedback. All modifications are highlighted in blue in the revised manuscript for easy identification of the changes made in response to your comments.

---

### Decision · Action_Editor_u7hG · 2024-09-22

**Recommendation:** Reject

**Comment:**

This is a hard case. On one hand, the paper studies a new setting and proposes a reasonable algorithm, which is appreciated by reviewers. On the other hand, the reviewers raised several critical concerns, at least two of which remain unaddressed. The first is the gap in the reported and published PPO results. The authors were appreciated for the investigation of what might contribute to the gap, yet it remains a question whether the baseline is strong enough to demonstrate the true value of the proposed algorithm. Second, the new setting's motivation is not convincing, despite multiple exchanges about motivating examples. After a closer look at the paper and discussions, the AE shares similar concerns with the motivation, unable to find a convincing scenario where the setting is a natural fit. Given these, the AE feels the work would benefit from major revisions before getting accepted.

**Audience:**

The paper is potentially interesting to the MARL (multi-agent RL) community, especially those working on cooperative games.

**Claims And Evidence:**

This work is concerned with a MARL setting, where the agents want to cooperate while preserving their privacy. It proposes an algorithm Anticipation Sharing, provides an bound on policy return improvement (based on advantages), and tests the algorithm on several cooperative MARL tasks. The technical claims are supported by theory and numerical experiments, although there remain concerns with the gap between reported and published results of the PPO baseline. The claim on motivation of the problem setting, however, does not find strong support.

**Resubmission Of Major Revision:**

The authors may consider submitting a major revision at a later time.